# Metabolic fluxes for nutritional flexibility of *Mycobacterium tuberculosis*

Khushboo Borah[1] , Tom A Mendum[1] , Nathaniel D Hawkins[2] , Jane L Ward[2], Michael H Beale[2],
Gerald Larrouy-Maumus[3] , Apoorva Bhatt[4] , Martine Moulin[5,6], Michael Haertlein[5,6],
Gernot Strohmeier[7,8], Harald Pichler[7,8,9], V Trevor Forsyth[5,6,10], Stephan Noack[11] ,
Celia W Goulding[12] , Johnjoe McFadden[1] & Dany J V Beste[1,*]

## Abstract

The co-catabolism of multiple host-derived carbon substrates is required by *Mycobacterium tuberculosis* (Mtb) to successfully sustain a tuberculosis infection. However, the metabolic plasticity of this pathogen and the complexity of the metabolic networks present a major obstacle in identifying those nodes most amenable to therapeutic interventions. It is therefore critical that we define the metabolic phenotypes of Mtb in different conditions. We applied metabolic flux analysis using stable isotopes and lipid fingerprinting to investigate the metabolic network of Mtb growing slowly in our steady-state chemostat system. We demonstrate that Mtb efficiently co-metabolises either cholesterol or glycerol, in combination with two-carbon generating substrates without any compartmentalisation of metabolism. We discovered that partitioning of flux between the TCA cycle and the glyoxylate shunt combined with a reversible methyl citrate cycle is the critical metabolic nodes which underlie the nutritional flexibility of Mtb. These findings provide novel insights into the metabolic architecture that affords adaptability of bacteria to divergent carbon substrates and expand our fundamental knowledge about the methyl citrate cycle and the glyoxylate shunt.

**Keywords** chemostat; metabolic flux; metabolism; *Mycobacterium tuberculosis*; tuberculosis
**Subject Categories** Metabolism; Microbiology, Virology & Host Pathogen Interaction
**Mol Syst Biol. (2021) 17: e10280**

## Introduction

*Mycobacterium tuberculosis* (Mtb) is the causative agent of a global tuberculosis (TB) pandemic which has now reached staggering levels making Mtb once again a leading cause of death globally (World Health Organisation, 2020). The terrifying trend of increasing antibiotic-resistant TB is destabilising TB control measures making new therapeutics which target drug-resistant strains of Mtb an urgent priority (Singh *et al*, 2020). Mtb is an unusual bacterial pathogen, which has the remarkable ability to cause both acute life-threatening disease and clinically latent infections that can persist for the lifetime of the human host.

*Mycobacterium tuberculosis* spends much of its life cycle growing intracellularly within the phagosomal compartment of macrophages where nutrient availability will fluctuate (Warner, 2014; Huang *et al*, 2018). In addition to replicating within the phagosomal compartment, it has been shown that Mtb can escape the intracellular environment to survive extracellularly (Grosset, 2003), in other cell types and within the diverse and dynamic microenvironments of granulomas (Bussi & Gutierrez, 2019). Mtb must therefore be able to survive in a plethora of different microenvironments and nutrients. Experimental evidence has identified central carbon metabolism as instrumental to this pathogenic strategy, and therefore, our research is focused on investigating the metabolic capabilities of Mtb both *in vitro* and *ex vivo* (Beste *et al*, 2007a; Beste *et al*, 2011; Beste *et al*, 2013; Lofthouse *et al*, 2013; Basu *et al*, 2018; Borah *et al*, 2019; López-Agudelo *et al*, 2020).

*Mycobacterium tuberculosis* maintains a functional tricarboxylic acid (TCA) cycle, pentose phosphate pathway (PPP) and Embden–Meyerhof–Parnas (EMP) pathway, as well as enzymes providing a

---

1   Department of Microbial and Cellular Sciences, Faculty of Health and Medical Sciences, University of Surrey, Guildford, UK
2   Department of Computational and Analytical Sciences, Rothamsted Research, Harpenden, UK
3   MRC Centre for Molecular Bacteriology and Infection, Department of Life Sciences, Faculty of Natural Sciences, Imperial College London, London, UK
4   School of Biosciences, University of Birmingham, Edgbaston, UK
5   Life Sciences Group, Institut Laue-Langevin, Grenoble Cedex 9, France
6   Partnership for Structural Biology, Grenoble Cedex 9, France
7   Austrian Centre of Industrial Biotechnology, Graz, Austria
8   Institute of Organic Chemistry, NAWI Graz, Graz University of Technology, Graz, Austria
9   Institute of Molecular Biotechnology, NAWI Graz, BioTechMed Graz, Graz University of Technology, Graz, Austria
10  Faculty of Natural Sciences, Keele University, Staffordshire, UK
11  Institute of Bio- and Geosciences 1: Biotechnology 2, Forschungszentrum Jülich GmbH, Jülich, Germany
12  Department of Pharmaceutical Sciences & Molecular Biology & Biochemistry, University of California Irvine, Irvine, CA, USA
    *Corresponding author. Tel: +44 1483 686785; E-mail: d.beste@surrey.ac.uk

metabolic link between glycolysis and the TCA cycle (Beste & McFadden, 2010). Mtb also has two alternative pathways (methyl citrate cycle and the B12-dependent methylmalonyl pathway) for metabolising propionyl-CoA derived from the metabolism of sterols, uneven branched chain fatty acids and amino acids (Eoh & Rhee, 2014). These pathways allow Mtb to utilise a wide range of carbon sources that includes carbohydrates, sugars, fatty acids, amino acids and sterols (Warner, 2014). Whilst the basic architecture of central carbon metabolism of Mtb is now well established, there are still many questions regarding how the flux of metabolites through this network is modulated under various different nutritional conditions.

Using stable isotope-labelled nutrients for studying the metabolism of Mtb has proved extremely informative (de Carvalho et al, 2010; Marrero et al, 2010; Beste et al, 2011; Beste et al, 2013; Eoh & Rhee, 2014; Nandakumar et al, 2014; Basu et al, 2018; Borah et al, 2019). Metabolic labelling experiments using $^{13}$C-labelled combinations of acetate, glycerol and glucose have demonstrated that Mtb is able to co-catabolise two carbon sources simultaneously demonstrating that Mtb does not use carbon catabolite repression to regulate metabolism (de Carvalho et al, 2010). This work also suggested that Mtb not only co-catabolised these carbon substrates but did so in a compartmentalised and segregated manner. However, importantly this work was not able to determine the metabolic flux profiles on these combinations of carbon substrates. Understanding metabolic fluxes of Mtb during co-catabolism of multiple carbon sources will allow us to identify nodes of metabolism most amenable to therapeutic intervention. By combining isotopomer labelling with our steady-state chemostat model system of mycobacterial growth allowed us to perform $^{13}$C-metabolic flux analysis (MFA) at different growth rates in carbon limited conditions (Beste et al, 2011). Previously, we identified the activity of a novel GAS pathway for pyruvate dissimulation when Mtb was growing slowly on glycerol and oleic acid and demonstrated that the pathway requires isocitrate lyase and the enzymes of the anaplerotic node (Beste et al, 2011) both of which are important for the survival of Mtb in the host (McKinney et al, 2000; Basu et al, 2018).

Glycerol was not considered an important carbon source for Mtb as glycerol kinase (glpK), which is an essential gene for the conversion of glycerol to glycerol 3-phosphate (Beste et al, 2009), is dispensable for the growth of Mtb in a murine TB model (Pethe et al, 2010). However, the detection of glycerol in human-like TB lesions has compelled a re-evaluation of the role of glycerol and glycerol containing metabolites in the life cycle of Mtb (Safi et al, 2019). Moreover, Mtb has been shown to co-metabolise a mixture of carbon substrates when growing in host macrophage cells that included an unknown glycolytic C3 substrate (Beste et al, 2013) that could potentially be glycerol.

Metabolomic and gene-deletion studies have highlighted fatty acids and cholesterol as critical to the nutrition, survival and virulence of Mtb (McKinney et al, 2000; Pandey & Sassetti, 2008) and that Mtb encounters and co-metabolises both substrates simultaneously in vivo (Wilburn et al, 2018). Cholesterol, uneven chain fatty acids and branched chain amino acids are all catabolised to provide Mtb with the metabolite propionyl-CoA which although required for the synthesis of important virulence cell wall lipids (phthiocerol dimycocerosate (PDIM), polyacylated trehalose and sulpholipids) is also toxic if allowed to accumulate intracellularly (Griffin et al, 2012; Lee et al, 2013). In addition to metabolising propionyl-CoA

through the methyl citrate cycle (MCC) or in the presence of vitamin B12, the methylmalonyl cycle, Mtb can also sequester propionyl-CoA into methyl branched cell wall lipids (Lee et al, 2013). During infection, it is thought that Mtb uses fatty acids to prime this process further suggesting that cholesterol and fatty acid metabolism occur simultaneously in vivo (Lee et al, 2013; Wilburn et al, 2018). Despite numerous biochemical studies to elucidate the biochemical degradation pathways and studies exploring the role of specific enzymes in cholesterol and fatty acid metabolism, the metabolic flux profile of Mtb growing on this combination of substrates has never been directly measured. Therefore, in this study we performed $^{13}$C-MFA on steady-state, slowly growing cultures of Mtb using an extended version of our $^{13}$C isotopomer model (Beste et al, 2011), which includes the MCC. We compared the metabolic flux profile of chemostat cultures of Mtb growing in defined carbon limited conditions growing with cholesterol/acetate with those growing on glycerol/Tween 80 to reflect carbon sources available to Mtb during the life cycle of this pathogen within the host.

We demonstrate that when Mtb is growing slowly on cholesterol and acetate, Mtb utilises a complete TCA cycle in combination with the glyoxylate shunt. There is very little demand for the MCC in these conditions as lipid fingerprinting identified that the propionyl-CoA-derived cholesterol is being preferentially incorporated into lipids. Conversely when growing on glycerol and Tween 80, Mtb utilises an incomplete TCA cycle and the methyl citrate cycle is reversed to provide propionyl-CoA for the synthesis of virulence lipids. This work highlights that re-routing fluxes through the TCA cycle, MCC and the glyoxylate shunt and in particular the ability to alternate the direction of the MCC, whilst co-metabolising carbon substrates during slow growth is critical to the metabolic flexibility of Mtb.

## Results

### Isotopic profiling of chemostat grown Mtb demonstrated efficient co-catabolism of carbon sources without metabolic compartmentalisation

To define the metabolic profile of Mtb growing on cholesterol and acetate, Mtb H37Rv cultures were grown in carbon limited chemostats operating at a dilution rate of $\mu$ = 0.01 h (doubling time $(t_d)$ = 69 h). We adopted this dilution rate as we have previously demonstrated that the transcriptional response at this growth rate has many similarities to the transcriptional response characteristic of the adaptation of Mtb to the macrophage environment, and others have shown a similar profile from Mtb isolated from sputum (Beste et al, 2007b). The cultures were grown in Roisin's minimal medium (Beste et al, 2005) with a combination of either cholesterol/acetate (CHL-ACE) or glycerol/Tween 80 (GLY-OLA) as previously described (Beste et al, 2011). After approximately three volume changes, little variation was observed in the $CO_2$ and biomass production rates, indicating that a metabolic steady-state conditions had been attained (Fig EV1). Data from the chemostat cultures demonstrated that Mtb was able to co-metabolise acetate and cholesterol simultaneously with similar yields of bacteria to cultures growing on glycerol and Tween 80. Substrate uptake rates were also comparable under the two growth conditions, but the $CO_2$ production rate was significantly higher in CHL-ACE (Table 1).

**Table 1.** Steady-state characteristics of *Mycobacterium tuberculosis* grown in carbon limited chemostats.

| Carbon sources | CHL-ACE | GLY-OLA |
|---|---|---|
| Dilution rate (h$^{-1}$) | 0.01 | 0.01 |
| Biomass (g dry weight l$^{-1}$) | 0.49 | 2.18 |
| CFU (× 10$^7$ ml$^{-1}$) | 3.9 | 33 |
| Cell weight (pg dry weight cfu$^{-1}$) | 12.5 | 6.7 |
| Substrate consumption rate (mmol g biomass$^{-1}$ h$^{-1}$) | Acetate = 0.26 | Glycerol = 0.23 |
| | Cholesterol = 0.0085 | Oleic acid = 0.002 |
| CO$_2$ production rate (mmol g biomass$^{-1}$ h$^{-1}$) | 0.245 | 0.178 |
| Yield (g biomass$^{-1}$ mmol carbon$^{-1}$) | 0.013 | 0.012 |

For the labelling experiments, steady-state chemostat cultures were switched to identical media containing $^{13}$C-labelled substrates (30% [$^{13}$C$_3$]glycerol OR 100% [$^{13}$C$_2$]acetate). Samples were taken every volume change for a total of four volume changes to ensure that an isotopic stationary state was attained. Labelling of proteinogenic amino acids and intracellular metabolites was measured using gas chromatography–mass spectrometry (GC-MS) or liquid chromatography–mass spectrometry (LC-MS) as previously described (Beste *et al,* 2011). Proteinogenic amino acids are commonly measured for $^{13}$C-MFA as they are much more abundant and stable than their precursors and provide extensive labelling information. Moreover, amino acids can be used to directly deduce the labelling patterns of their precursor metabolites. The labelling pattern of the amino acids changed very little between the third and fourth volume change confirming that the culture had reached an isotopic steady state which is essential for determining intracellular fluxes using $^{13}$C-MFA (Table 1; Fig EV1C and D).

Previous studies indicated that Mtb was operating a form of compartmentalised metabolism when growing on solid 7H10 agar containing combinations of the carbon sources glucose, glycerol and acetate whereby individual substrates had distinct metabolic fates (de Carvalho *et al,* 2010). However, our $^{13}$C labelling data from chemostat cultures at metabolic and isotopic steady state showed no evidence of such compartmentalisation when Mtb was grown with either GLY-OLA or CHL-ACE as evidenced by uniform distribution of both unlabelled and $^{13}$C-labelled substrates (Fig 1).

For steady-state CHL-ACE Mtb cultures, the $^{13}$C labelling profiles reflect the entry points of cholesterol into central carbon metabolism. Cholesterol catabolism yields four propionyl-CoA, four acetyl-CoA, one pyruvate and one succinyl-CoA of which succinyl-CoA, pyruvate and acetyl-CoA enter central carbon metabolism directly (Crowe *et al,* 2017). Propionyl-CoA is toxic to Mtb and can be metabolised by either: (i) the methyl citrate cycle to pyruvate, (ii) the B12-dependent methylmalonyl pathway leading to succinyl-CoA, or (iii) used in cell wall lipogenesis. In our experiments, the methylmalonyl pathway is not active (as Roisin's media lacks vitamin B12).

For the CHL-ACE experiments, the labelling profile of succinate (SUC), pyruvate (PYR) and the pyruvate-derived amino acids

(alanine (ALA), valine (VAL) and leucine (LEU)) had ≥ 50% unlabelled carbon indicating that the carbon backbone of these metabolites was predominantly derived from unlabelled cholesterol. This is expected as cholesterol enters central carbon metabolism as succinate and pyruvate. Canonical $^{13}$C labelling patterns were measured in the other metabolites reflecting that 60% of the total carbon was derived from $^{13}$C-labelled acetate and the remainder derived from unlabelled cholesterol indicating that metabolism was also not compartmentalised in these conditions (Fig 1). For GLY-OLA grown cultures, the labelling profiles were consistent across the different metabolites analysed; the backbone of these metabolites was synthesised primarily from glycerol, demonstrating that metabolism of GLY-OLA was also not compartmentalised.

## $^{13}$C isotopologue analysis of amino acids defines their biosynthetic routes

We performed $^{13}$C isotopologue analysis of proteinogenic amino acids. As expected, the profiles of all metabolites derived from CHL-ACE were different to that derived from GLY-OLA grown Mtb (Fig 2). The labelling profiles of amino acids reflect their biosynthetic origin for both GLY-OLA and CHL-ACE grown Mtb. For the GLY-OLA cultures, the labelling patterns of aspartate (ASP/N), threonine (THR), isoleucine (ILE), lysine (LYS) and methionine (MET) were similar, consistent with a common biosynthetic origin of these amino acids from oxaloacetate. The isotopologue composition of tyrosine (TYR) and phenylalanine (PHE) was also alike reflecting their common precursors – phosphoenolpyruvate and erythrose-4-phosphate. Similarly, ornithine (ORN) and glutamate (GLU/N), which are both derived from α-ketoglutarate, had similar labelling patterns. The isotopologue profiles of proteinogenic amino acids derived from steady-state cultures grown in CHL-ACE also reflect their biosynthetic origin. THR and MET had near identical profiles with ASP/N with M + 2 and M + 4 isotopomers having the highest $^{13}$C labelling, indicating their synthesis from ASP. GLU/N and ORN, TYR and PHE and ALA and SER all had very similar profiles indicating the common biosynthetic precursor for each of these pairs of amino acids. Whilst manual analysis of $^{13}$C isotopologue data allows qualitative conclusions, systems level analysis is required in order to quantitate the metabolic fluxes.

## Improved $^{13}$C-Metabolic Flux Analysis using an expanded isotopomer model

$^{13}$C-MFA is the preferred tool for quantitative characterisation of metabolic phenotypes in steady-state cultures (Wiechert, 2001). Previously, we developed an isotopomer model of Mtb's central carbon metabolism comprising of the TCA cycle, glycolysis, pentose phosphate pathway (PPP) and anaplerotic reactions, which allowed us to analyse $^{13}$C-isotopomer data and compute metabolic fluxes (Beste *et al,* 2011). For this study, we have expanded our capacity for predicting metabolic fluxes by adding the methyl citrate cycle (MCC) and amino acid degradation pathways to our previous $^{13}$C-isotopomer model (Table EV1). Absolute flux distributions were derived from the two steady-state chemostat conditions (Fig 3) using the INCA platform (Young, 2014). In our previous study, we were unable to determine the flux values unambiguously for Mtb growing in GLY-OLA. Here by using our extended isotopomer

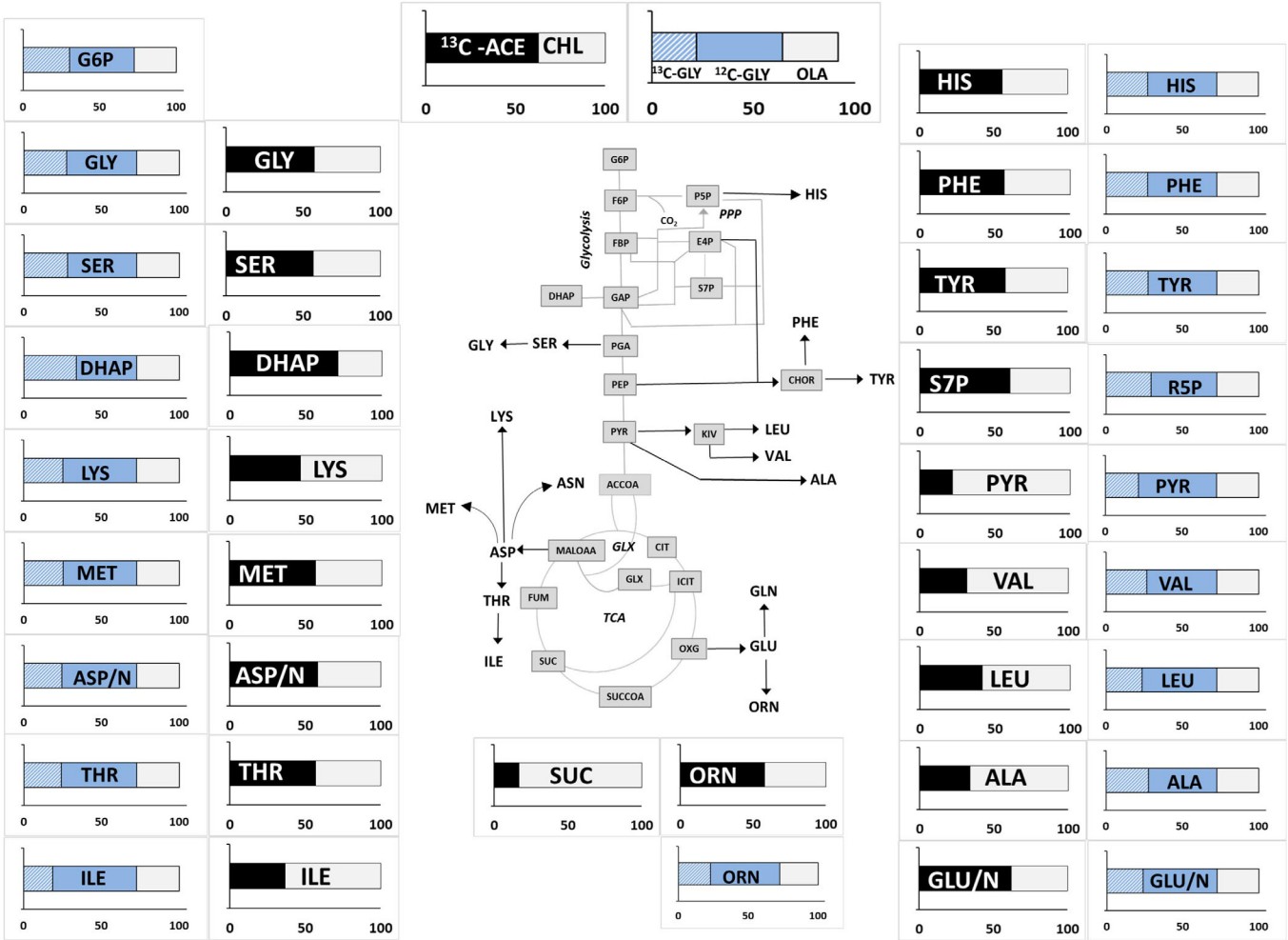

**Figure 1.** **$^{13}$C incorporation into the proteinogenic amino acids and intracellular metabolites from metabolic and isotopic steady-state chemostat Mtb cultures.**

Label distribution is shown in metabolites from Mtb grown in 30% [$^{13}C_3$] glycerol and unlabelled Tween 80 which provides Mtb with oleic acid (GLY-OLA) and 100% [$^{13}C_2$] acetate and unlabelled cholesterol (CHL-ACE). Average $^{13}$C incorporation was calculated for metabolites harvested at a steady-state growth and are shown as the amount labelled ($^{13}$C) and unlabelled ($^{12}$C) using 3–4 independent measurements. Data are average of 3 replicate measurements. DHAP (dihydroxyacetone phosphate), PYR (pyruvate) ALA (alanine), GLY (glycine), SER (serine), LYS (lysine), MET (methionine), ASP/N (aspartate/asparagine), THR (threonine), ILE (isoleucine), ORN (ornithine), GLU/N (glutamate/glutamine), SUC (succinate), LEU (leucine), ILE (isoleucine), VAL (valine), PHE (phenylalanine), TYR (tyrosine), HIS (histidine), S7P (sedoheptulose-7-phosphate), measured for CHL-ACE and R5P (ribose 5-phosphate) and measured for GLY-OLA cultures, are plotted on a metabolic map showing reactions for glycolysis, PPP, GLX (glyoxylate shunt) and the TCA cycle. Aspartate/asparagine and glutamate/glutamine pools are lumped as both asparagine and glutamine were reduced to aspartate and glutamate, respectively, during acid hydrolysis.

model, we obtained a unique metabolic flux profile for these conditions building on our previous work (Fig 3). A comparison of the current and the previous metabolic flux profiles of Mtb growing in GLY-OLA in steady-state cultures (Fig EV2) illustrates the similarities between the solutions and shows that extending the isotopomer model has improved our ability to resolve the metabolic fluxes of Mtb.

**Distinct metabolic flux partitioning between the TCA cycle and glyoxylate shunt during co-catabolism**

The resolved flux distributions of Mtb growing on CHL-ACE demonstrated that there is a significant difference between how Mtb partitions carbon flux between the TCA cycle and the glyoxylate shunt in CHL-ACE as compared with GLY-OLA (Fig 3A). When growing on CHL-ACE, Mtb operates a complete oxidative TCA cycle in combination with an active glyoxylate shunt. There is only a very low flux through pyruvate dehydrogenase (PDH) (TCA R1) with the small amount of cholesterol-derived pyruvate being channelled to leucine, valine and alanine biosynthesis (Figs 3A, and 4A and B). In concordance with our previous work, Mtb uses an incomplete TCA cycle when growing on GLY-OLA (Beste *et al*, 2011).

Determining the individual anaplerotic fluxes from $^{13}$C-isotopomer data presents a major challenge. For example, it is not possible to discriminate between the reactions of malic enzyme (MEZ) and pyruvate carboxylase (PCA) with high confidence so the

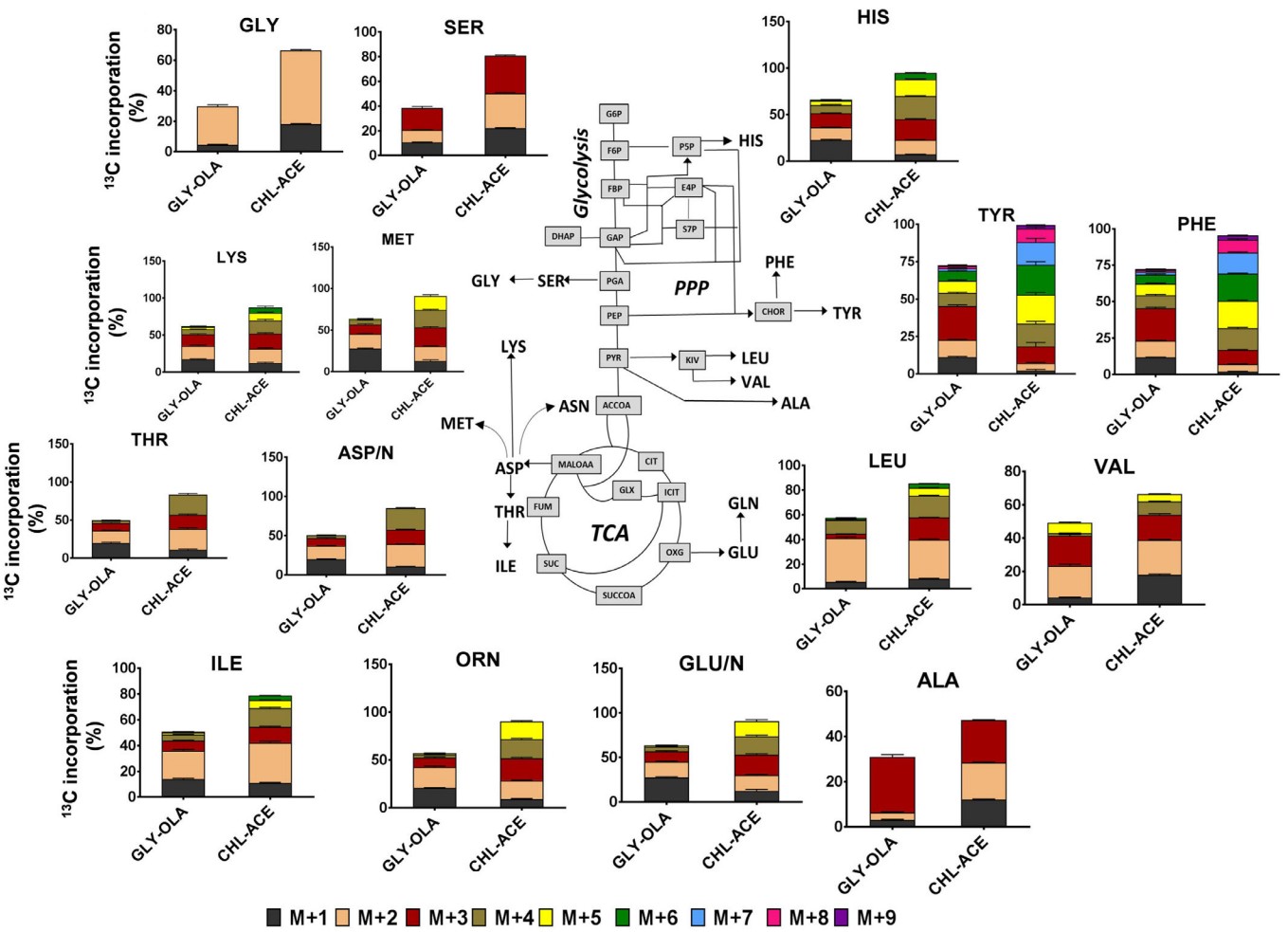

**Figure 2.** ¹³C isotopologue analysis of proteinogenic amino acids from Mtb grown on CHL-ACE and GLY-OLA.

Measurements were obtained from Mtb grown under steady-state chemostat conditions. ¹³C incorporation is shown for the mass isotopomers, which are labelled as ¹³C₁, ¹³C₂, etc. Measurements are shown for alanine (ALA), leucine (LEU), valine (VAL), glycine (GLY), serine (SER), methionine (MET), histidine (HIS), aspartate–asparagine (ASP/N), threonine (THR), lysine (LYS), isoleucine (ILE), glutamate–glutamine (GLU/N), ornithine (ORN), tyrosine (TYR) and phenylalanine (PHE). Glycolysis, pentose phosphate pathway (PPP), glyoxylate shunt (GLX) and the TCA cycle are depicted. Values are shown as mean ± error bars (s.d) of 3–4 replicates.

summed flux is shown between pyruvate and malate/oxaloacetate (ANA R2, Fig 4A and B). From the data, we identified that Mtb uses phosphoenolpyruvate carboxykinase (PEPCK or ANA R1) gluco-neogenically during growth on CHL-ACE to generate PEP, which is in accordance with the use of the glyoxylate shunt for replenishing the TCA intermediates, whereas, when growing on GLY-OLA, PEPCK operates in the carboxylating direction to fix $CO_2$ as we have described previously (Beste *et al*, 2011). The overall flux through ANA R2 is very low and not significantly different when growing on CHL-ACE or GLY-OLA.

Perhaps, surprisingly, there was no flux from pyruvate to PEP, the reaction catalysed by the fourth member of the anaplerotic node, pyruvate phosphate dikinase (PPDK), in either condition. There was flux towards pyruvate mediated by pyruvate kinase (PYK = EMP R6) and this was higher on GLY/OAA than with CHL/ACE (Fig 4C and D). We previously demonstrated using mutagenesis that Rv1127c (annotated as PPDK) was essential for the growth of Mtb in rich media containing cholesterol even in the presence of an

additional growth permissive carbon source (including acetate) (Basu *et al*, 2018). This phenotype was associated with the inability of the knockout Mtb strain to metabolise the cholesterol by-product propionate, which is toxic to bacterial cells. High-throughput Tn screen also identified Rv1127c as required for growth on cholesterol (Griffin *et al*, 2011). However, other studies suggest that Mtb lacks PPDK activity (Rhee *et al*, 2011). In order to explore this further, we performed a ¹³C labelling experiment growing Mtb *in vitro* on Roisin's minimal medium containing [3, 4-¹³C₂] cholesterol as a sole carbon source (Fig 4E) This experiment identified [¹³C₂] pyruvate, as expected when cholesterol is catabolised; however, all of the PEP detected was only labelled in one carbon (M + 1), indicating that the other ¹³C label has been lost as $CO_2$ via the TCA cycle consistent with the metabolic flux prediction that Mtb lacks PPDK activity (Fig 4D and E).

We observed only a very low gluconeogenic flux (Fig 4C and D) during growth on CHL-ACE, which was in sharp contrast to the glycolytic fluxes from glycerol to pyruvate, when grown on

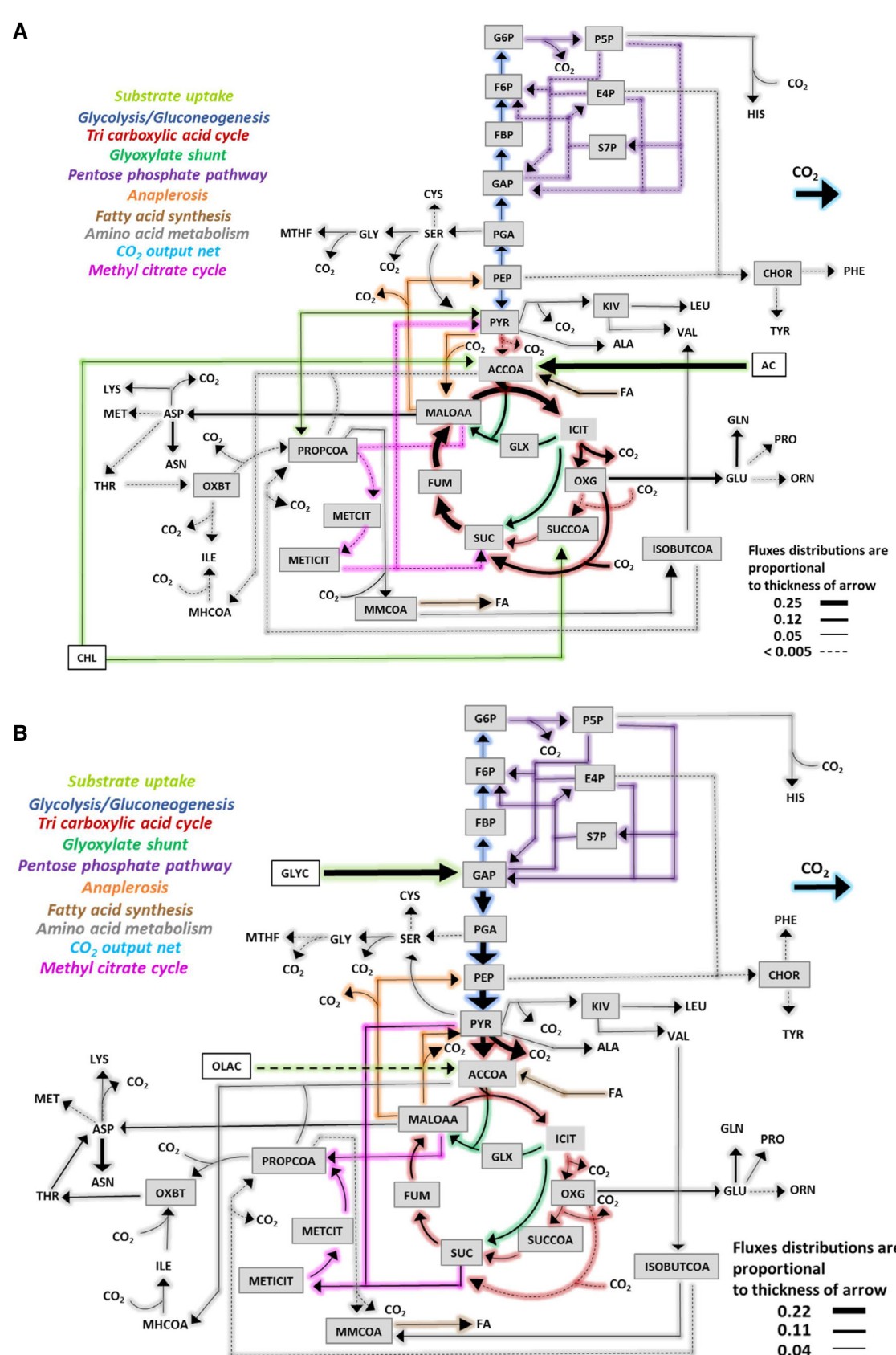

**Figure 3.**

**Figure 3. Flux distributions of Mtb grown on CHL-ACE (A) and GLY-OLA (B) carbon substrate combinations.**

A, B Flux distributions were calculated for the extended Mtb metabolic network shown in Table EV1. Fluxes are absolute values expressed as mmol g$^{-1}$ biomass h$^{-1}$.
$^{13}$C-labelled substrate consumption – acetate (ACE) and glycerol (GLYC) – and $CO_2$ production fluxes were fixed for the calculations. Steady-state $^{13}$C mass isotopomer measurements for the protein-derived amino acids from CHL-ACE and GLY-OLA grown Mtb were used for flux estimations. Estimated fluxes in File EV3 are represented by arrows in Fig 3A and B. The width of each line is proportional to the underlying flux value. Metabolite lists for the flux maps are included in supplementary data file EV3.

GLY-OLA. Pyruvate kinase (PYK) was however active under both conditions, where it funnels the PEP generated from PEPCK to pyruvate when growing on CHL-ACE (Fig 4C and D). On GLY-OLA, the flux through PYK was higher, a prediction which agrees with reports that this enzyme is required for carbon co-metabolism (Noy *et al*, 2016). Fluxes through the pentose phosphate pathway (PPP) were much lower when Mtb was grown with CHL-ACE as compared to GLY-OLA. The PPP not only supplies pentose phosphates, but also reducing power in the form of NADPH which can then be used for lipid biosynthesis and other anabolic processes. This may reflect a higher demand for NADPH from the PPP when growing Mtb is growing on GLY-OLA as compared to cultures containing the highly reduced carbon source, cholesterol (Fig 4C and D).

**Methyl citrate cycle fluxes are reversed during co-metabolism of glycerol-oleic acid**

The MCC in mycobacteria is required for the utilisation of the propionyl-CoA derived from sterols, branched chain amino and fatty acids that are acquired from the host (Fig 5A). Here, we show that when Mtb is growing on cholesterol and acetate, there is very low flux through the MCC suggesting that this pathway is minimally required for cholesterol metabolism in the presence of acetate (Fig 5A and B). Previous studies have demonstrated that the necessity for the MCC can be overridden by supplying Mtb with AcCoA generating carbon sources which primes the conversion of propionyl-CoA into methylmalonyl-CoA-lipids such as sulpholipid-1 (SL-1), polyacyltrehalose (PAT) and phthiocerol dimycocerosate (PDIM) (Lee *et al*, 2013). This lipid synthesis may account for the increased biomass observed in CHL/ACE grown cells (Table 1). By contrast when Mtb is growing on GLY-OLA, there is a reversal of flux through the MCC (MCC R1-R3) presumably to generate propionyl-CoA for lipid biosynthesis, which is also supplemented by the degradation of the branched chain amino acid valine (D R1 and D R2) (Fig 5B–D).

In order to explore the incorporation of carbon into lipids, we performed MALDI-TOF analysis on Mtb from our steady-state chemostat cultures (Fig 6A–D). These data cannot be incorporated into flux analysis as there is no one-to-one mapping between carbon atoms in lipids and precursor metabolites. We identified a large number of complex lipid species, including phosphatidylinositol mannosides (PIMs) and SL-1 (Fig 6A–D) In order to test whether this was due to incorporation of $^{13}$C acetate into SL-1 or an increased chain length due to the incorporation of carbon from unlabelled cholesterol, as predicted by our flux profiles, we also measured the lipid fingerprint of control samples derived from the same chemostat cultures but prior to attaining an isotopic steady state. In accordance with our metabolic flux predictions, we identified an increase in chain length in SL-1 in these unlabelled control samples which confirmed our expectations that lipids derived from

CHL-ACE-grown Mtb had significant differences in the composition of the methylmalonyl-CoA-derived SL-1s, specifically an increase in chain length and abundance when compared to GLY-OLA grown Mtb (Fig 6A and B). This is concordance with other studies which show an increased chain length of SL-1 during growth of Mtb on cholesterol and propionate (Griffin *et al*, 2012). We also found significant mass shifts in triacylphosphatidylinositol dimannosides (AcPIMs) isolated from the chemostat samples. However, these shifts were not observed in the control samples indicating that in this case, the increased mass was due to $^{13}$C-labelled acetate or glycerol being incorporated into these lipids (Fig 6C and D). For CHL-ACE-grown Mtb, the observed increased mass shift (Fig EV3A and B) was likely due to incorporation of [$^{13}$C$_2$] acetate directly into these lipids in contrast to the GLY-OLA chemostat grown Mtb where unlabelled OLA was being predominantly used for acetylation (Fig EV3).

**Mtb utilises carbon and hydrogen from cholesterol to synthesise amino acids**

Since the synthesis of amino acids is required for biomass production, we also calculated the amino acid production rates. $^{13}$C-MFA demonstrated that there were significant differences in the amino acid biosynthesis fluxes between the two chemostat conditions. For example, the synthesis of ASP/ASN, GLU/GLN and ILE was significantly higher for Mtb growing slowly with CHL-ACE than when the carbon sources were GLY-OLA (Fig 7A). Scrutinising these data further shows that the fluxes through GLU/GLN synthesis ultimately contribute towards asparagine biosynthesis. This is in concordance with the data showing that asparagine accumulates during metabolism of the cholesterol by-product propionate (Lee *et al*, 2018).

The increase in biosynthesis of ILE during growth on CHL-ACE is of interest as in other fungi and bacteria it has been shown that branched chain amino acid biosynthesis can also function as a mechanism to dissipate reductants (Shimizu *et al*, 2010; Zhang *et al*, 2018). Supported by the lipid profiling data here, it is thought that Mtb primarily uses lipids as an electron sink to maintain redox homeostasis when growing on highly reduced carbon sources such as cholesterol (Singh *et al*, 2009; Mavi *et al*, 2019). However, the role of amino acids as potential electron sinks has never been explored for Mtb. Thus, we grew Mtb in Roisin's minimal medium containing uniformly deuterated cholesterol and measured the incorporation of deuterium into amino acids. This analysis showed that Mtb incorporates hydrogen from cholesterol into several amino acids with branched chain amino acids having the highest level of incorporation (Fig 7B). These data suggest that amino acid anabolism could provide Mtb with another mechanism for sinking electrons and maintaining redox homeostasis when consuming highly reduced carbon sources such as cholesterol. Interpretation of $^2$H

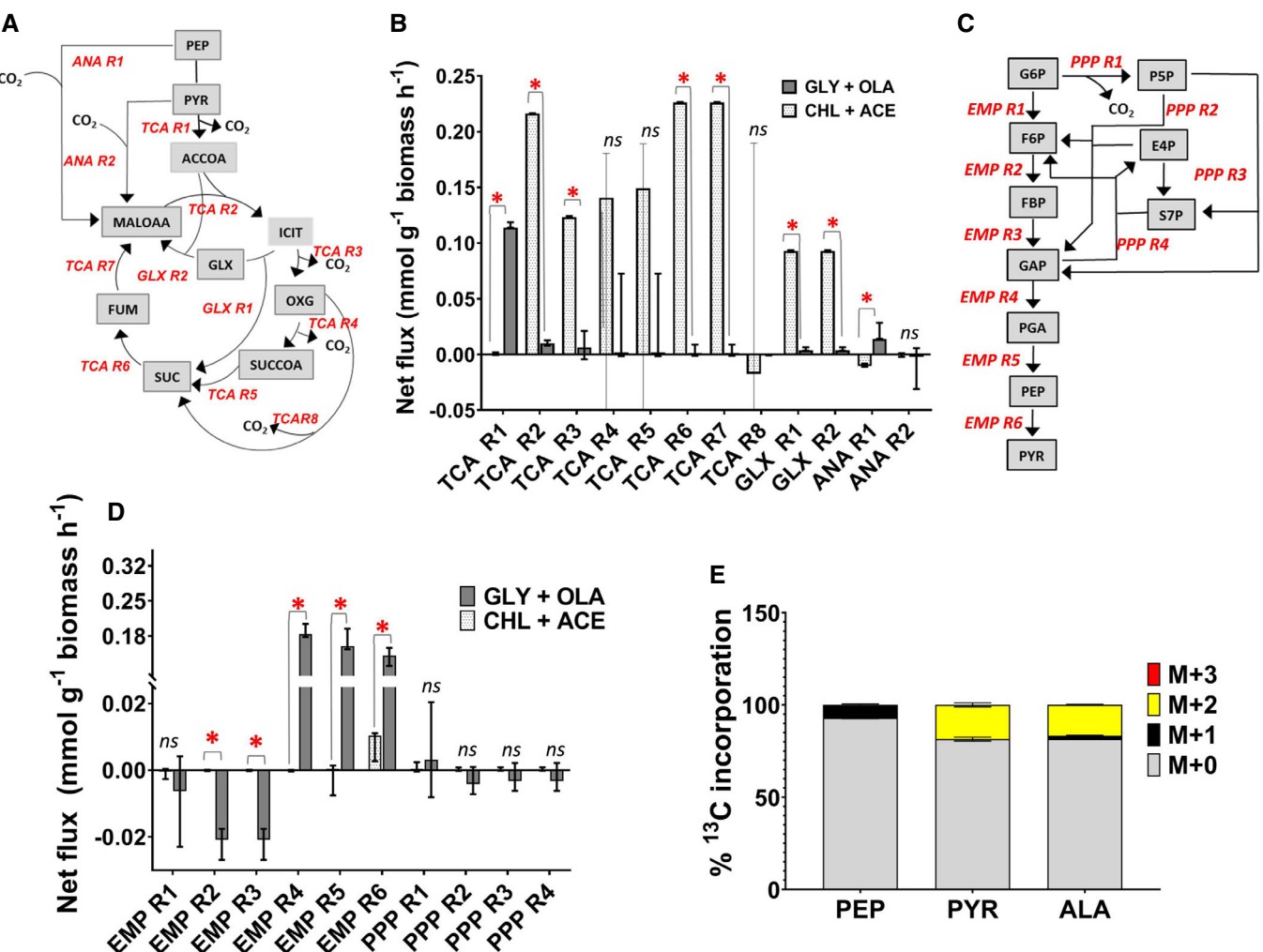

**Figure 4. Flux distributions through the TCA cycle, glyoxylate shunt, anaplerotic pathways, glycolysis and gluconeogenesis for Mtb grown on CHL-ACE and GLY-OLA.**

A   The network reactions for the TCA cycle (TCA R1, TCA R2, TCA R3, TCA R4, TCA R5, TCA R6, TCA R7), glyoxylate shunt (GLX R1, GLX R2) and anaplerotic reactions (ANA R1, ANA R2) along with their directionalities are shown.

B   Quantitative comparisons of the best-fit net fluxes derived from three chemostat cultivations and three technical replicates. The error bars on these measurements show the lower and upper 95% confidence limits calculated using parameter continuation in INCA to determine the sensitivity of the minimised sum of squared residuals as a function of the flux value. Two fluxes are considered significantly different if the confidence limits do not overlap ($P < 0.05$). Statistically significant differences for the fluxes measured on the two growth substrates are indicated by *; ns denotes not significant.

C   The network reactions for glycolysis (EMP R1, EMP R2, EMP R3, EMP R4, EMP R5, EMP R6) and for PPP (PPP R1, PPP R2, PPP R3, PPP R4) along with their directionalities.

D   The best-fit net fluxes for glycolysis and PPP, respectively, derived from three chemostat cultivations and three technical replicates. A negative flux indicated the reverse directionality of the flux to that shown in A and C. The error bars on these measurements show the lower and upper 95% confidence limits for the fluxes calculated using parameter continuation in INCA to determine the sensitivity of the minimised sum of squared residuals as a function of the flux value. Two fluxes are considered significantly different if the confidence limits do not overlap ($P < 0.05$) Statistically significant differences for the fluxes measured on the two growth substrates are indicated by *; ns denotes not significant.

E   $^{13}$C incorporation in pyruvate (PYR) and PEP of Mtb grown in [3, 4-$^{13}$C$_2$] cholesterol. Values are shown as mean $\pm$ s.e.m. of 3–4 biological replicates.

Data information: Metabolites shown in A and D are MALOAA (malate + oxaloacetate), SUC (succinate), ACCOA (acetyl coenzyme A), PYR (pyruvate), ICIT (isocitrate), GLX (glyoxylate), OXG (α-ketoglutarate), SUCCOA (succinyl coenzyme A), FUM (fumarate), G6P (glucose-6-phosphate), F6P (fructose-6-phosphate), FBP (fructose 1,6-bisphosphate), GAP (glyceraldehyde-3-phosphate), PGA (phosphoglyceric acid), PEP (phosphoenolpyruvate), PYR (pyruvate), P5P (pentose-5-phosphate), E4P (erythrose-4-phosphate) and S7P (sedoheptulose-7-phosphate).

labelling data is complicated by the presence of significant deuterium kinetic isotope effects (Liu *et al*, 2016). However, $^2$H tracers have proved invaluable in resolving redox metabolism in different organisms (Lewis Caroline *et al*, 2014; Jang *et al*, 2018).

## Discussion

*Mycobacterium tuberculosis* is known to be able to acquire and assimilate multiple carbon sources from the host during infection

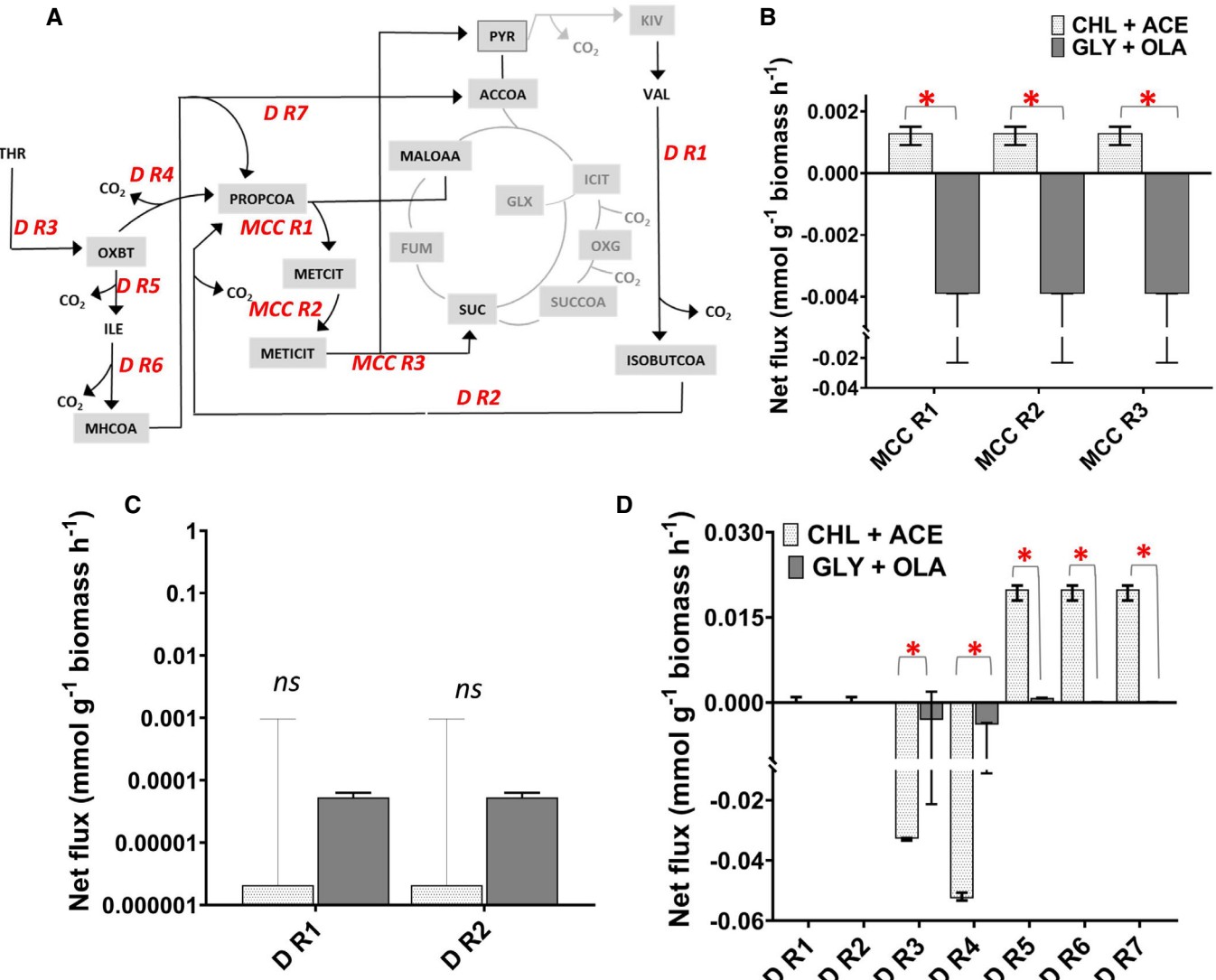

**Figure 5. Flux distributions through the methyl citrate cycle and amino acid degradation pathways of Mtb grown on CHL-ACE and GLY-OLA.**

A   The network reactions for methyl citrate cycle (MCC R1, MCC R2, MCC R3) and amino acid degradation pathways (D R1, D R2, D R3, D R4, D R5, D R6, D R7) along with their directionalities are shown.

B   Best-fit absolute net fluxes of the methyl citrate cycle derived from three chemostat cultivations and three technical replicates. The error bars are lower and upper 95% confidence limits of the net fluxes calculated using parameter continuation in INCA to determine the sensitivity of the minimised sum of squared residuals as a function of the flux value. Two fluxes are considered significantly different if the confidence limits do not overlap ($P < 0.05$). Statistically significant differences for the fluxes measured on the two growth substrates are indicated by *; ns denotes not significant.

C, D   Best-fit absolute net fluxes for valine degradation, threonine and isoleucine degradation on CHL-ACE and GLY-OLA derived from three chemostat cultivations and three technical replicates. A negative flux indicated the reverse directionality of the flux to that shown in A. The error bars on these measurements show the lower and upper 95% confidence limits for the fluxes calculated using parameter continuation in INCA to determine the sensitivity of the minimised sum of squared residuals as a function of the flux value. Two fluxes are considered significantly different if the confidence limits do not overlap ($P < 0.05$). Fluxes are significantly different if the confidence limits do not overlap. Statistically significant differences for the fluxes measured on the two growth substrates are indicated by *; ns denotes not significant.

Data information: Abbreviations for metabolites are MALOAA (malate + oxaloacetate), SUC (succinate), ACCOA (acetyl coenzyme A), PYR (pyruvate), METCIT (methyl citrate), METICIT (methyl isocitrate), MMCOA (methylmalonyl coenzyme A), ISOBUTCOA (isobutyl coenzyme A), OXBT (oxobutanoate), ILE (isoleucine), THR (threonine) and MHCOA (methylacetoacetyl coenzyme A).

including sugars, host-derived fatty acids, amino acids and cholesterol as sources of nitrogen, carbon and energy during host infection. This co-catabolism has been demonstrated to occur both *in vitro* and during intracellular growth (de Carvalho *et al*, 2010).

Details of the metabolic mechanisms involved in this process have been inferred from [13]C labelling experiments in which specific nutrients were observed to have defined fates suggesting some form of compartmentalisation of metabolites (de Carvalho *et al*, 2010).

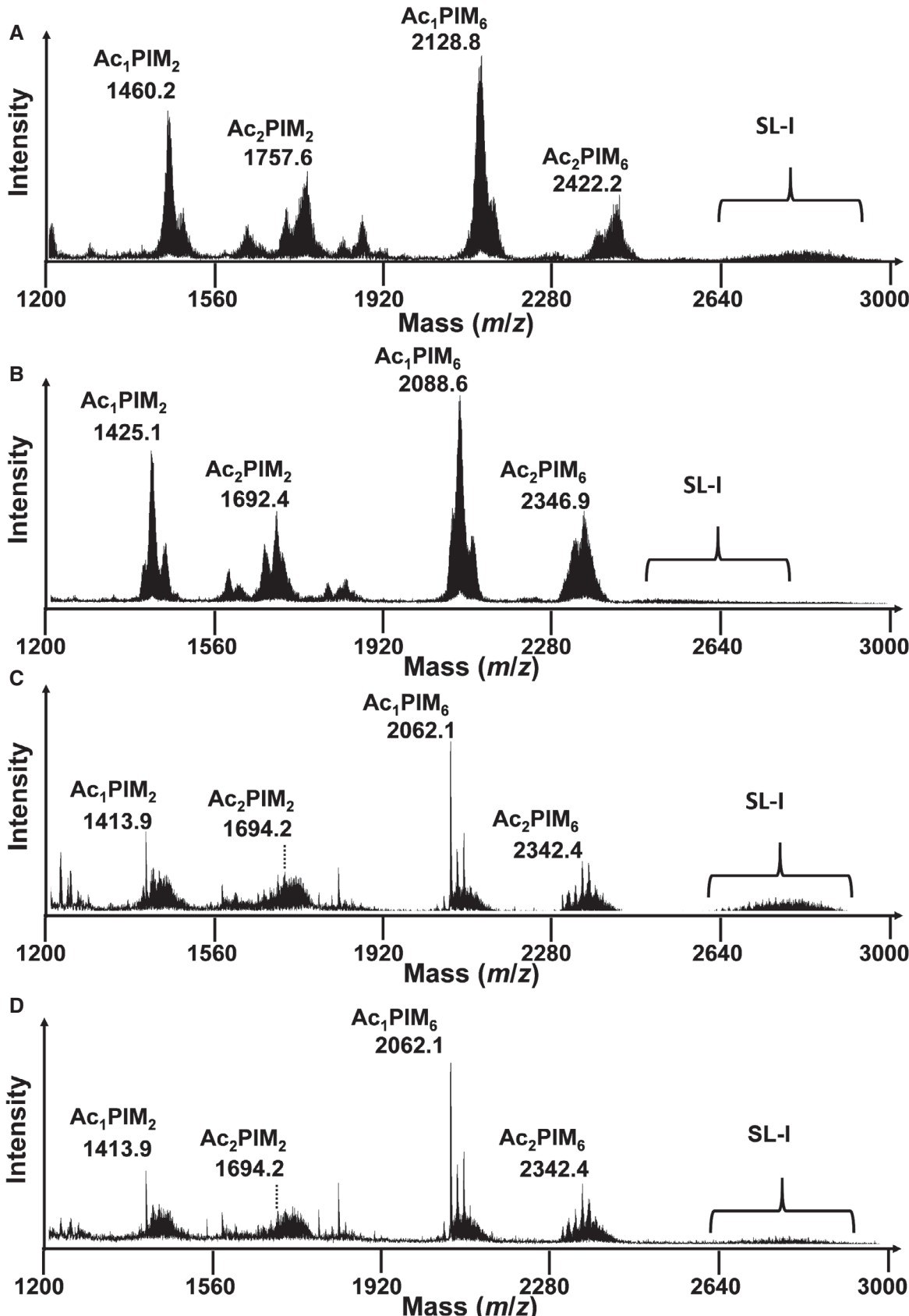

**Figure 6.**

**Figure 6. Negative ion mode MALDI-ToF mass spectra showing the lipid fingerprint of Mtb under different carbon sources in chemostat.**

A–D  Ac1PIM2: tri-acyl phosphatidylinositol mannoside di-mannose; Ac2PIM2: tetra-acyl phosphatidylinositol mannoside di-mannose; Ac1PIM6: tri-acylated phosphatidyl-myo-inositol hexamannoside; Ac2PIM6: tetra-acylated phosphatidyl-myo-inositol hexamannoside; and SL-I: sulpholipid. The ion at $m/z$ 1413.9 is assigned to the deprotonated PIM2 containing 2 C16:0 and 1 C19:0; the ion at $m/z$ 1,694.1 is assigned to the deprotonated PIM2 containing 2 C16:0 and 2 C19:0; the ion at $m/z$ 2,062.1 is assigned to the deprotonated PIM6 containing 2 C16:0 and 1 C19:0; the ion at $m/z$ 2,342.9 is assigned to the deprotonated PIM2 containing 2 C16:0 and 2 C19:0; and SL-I is observed as a broad set of peaks representing multiple lipoforms corresponding to differing numbers of $CH_2$ units. Multiple lipoforms of SL-1 ($m/z$ 2500 to $m/z$ 3000) are also indicated SL-I. Mass shifts due to $^{13}$C labelling of whole bacteria lipid fingerprint prepared from metabolic and isotopic steady-state chemostat Mtb grown with [$^{13}C_2$] acetate and unlabelled cholesterol (A), [$^{13}C_3$] glycerol and unlabelled Tween 80 (B), unlabelled acetate and cholesterol (C) and unlabelled glycerol and Tween 80 (D). Samples taken from the same chemostat at metabolic steady state (A) and (B) and isotopic in stationary state (C) and (D). The resolved mass shifts observed for Ac1PIM2 are shown to indicate significant lipid changes between the two conditions.

However, in our steady-state, slow-growing cultures we demonstrated efficient co-catabolism of either cholesterol and acetate, or glycerol and oleic acid but found no evidence of compartmentalised metabolism. This apparent discrepancy suggests that metabolic compartmentalisation may be carbon source-dependent or may reflect differences in experimental setup. The previous studies were performed using Mtb grown in non-metabolic steady-state conditions on Middlebrook 7H10 agar containing in addition to the carbon sources (one of which was $^{13}$C labelled) significant amounts of unlabelled glutamate which Mtb can use as both a nitrogen and a carbon source (Lofthouse *et al*, 2013). The instationary metabolic state of the culture would affect the rates of $^{13}$C incorporation into intermediates, whilst the additional unlabelled carbon source would have diluted out the label thus potentially confounding the analysis.

Compartmentalisation in prokaryotic cells can involve micro-compartments or cages. These are semi-permeable protein shells that sequester enzyme complexes from a segment of a metabolic pathway, but that are permeable to metabolites. These structures allow pathway reactions to be spatially organised and so can enhance flux through multi-step pathways or protect cells from toxic metabolic intermediates (Saier, 2013; Lau *et al*, 2018). To date, only relatively small cages have been identified in mycobacteria, which encapsulate pathways involved in oxidative stress and iron storage pathways (Contreras *et al*, 2014). Mtb does not encode the various subunits required to make-up the larger compartments that could, for example, encage whole metabolic pathways. Therefore, with our current knowledge, it is difficult to understand how Mtb could segregate metabolites from single carbon sources through a large pathway such as the TCA cycle, for example. The hypothesis that Mtb compartmentalises central carbon metabolism thereby requires further investigation.

The physiological role of Mtb's glyoxylate shunt is to supply and replenish the TCA and MCC cycle intermediates, oxaloacetate, to ensure adequate supply for gluconeogenesis and amino acid

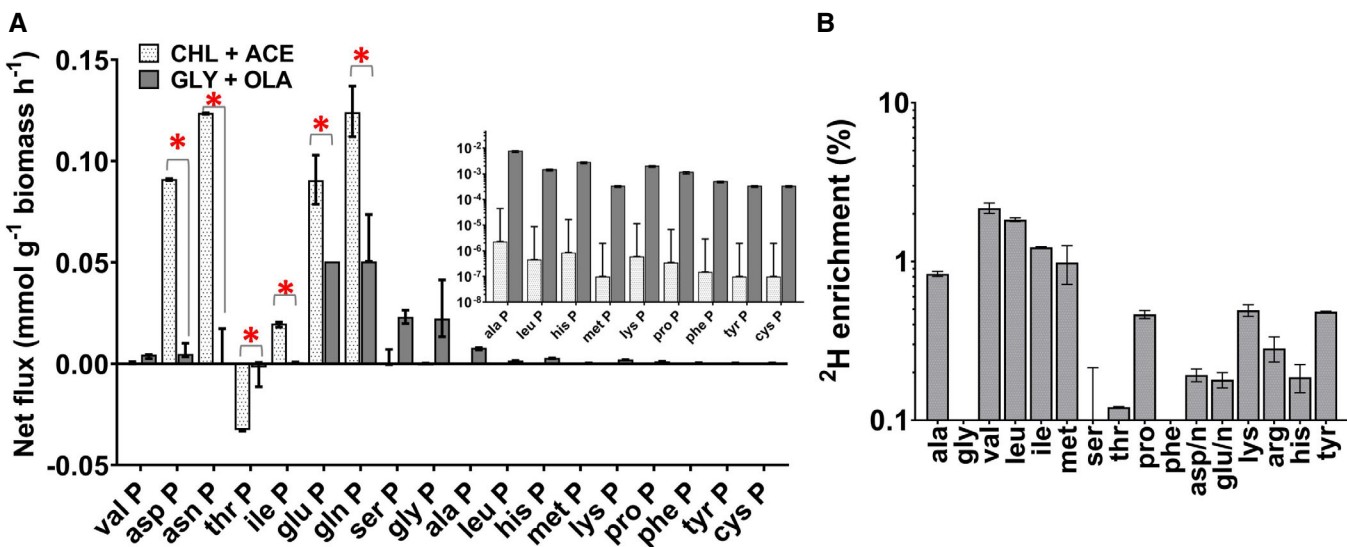

**Figure 7. Amino acid biosynthesis.**

A  Best-fit net fluxes derived from three chemostat cultivations and three technical replicates are shown for the synthesis of amino acids alanine (ala P), valine (val P), leucine (leu P), tyrosine (tyr P), phenylalanine (phe P), histidine (his P), glutamine (gln P), glutamate (glu P), proline (pro P), ornithine (orn P), asparagine (asn P), aspartate (asp P), threonine (thr P), isoleucine (ile P), methionine (met P), lysine (lys P), glycine (gly P), cysteine (cys P) and serine (ser P). The error bars on these measurements show the lower and upper 95% confidence limits for the fluxes calculated using parameter continuation in INCA to determine the sensitivity of the minimised sum of squared residuals as a function of the flux value. Two fluxes are considered significantly different if the confidence limits do not overlap ($P < 0.05$). Statistically significant differences for the fluxes measured on the two growth substrates are indicated by *.

B  $^2$H (deuterated hydrogen) incorporation in the amino acids of Mtb grown in minimal media with deuterated cholesterol as the sole carbon source. Values are mean ± s.e.m. (n = 3 biological replicates).

synthesis by bypassing the decarboxylating steps of the TCA cycle and thereby preserving carbon (Munoz-Elias & McKinney, 2005). The pathway is now known to have a multifaceted role in microbial growth and survival in a variety of conditions and is also critical for the virulence of a number of pathogens including Mtb (Dolan & Welch, 2018). During growth on CHL-ACE, Mtb had significantly higher fluxes through both the complete TCA cycle and the glyoxylate shunt than when grown with GLY-OLA.

In the highly oxygenated chemostat, cholesterol catabolism will generate reductants that need to be recycled (Mavi et al, 2019). Our flux results demonstrate that during slow growth with CHL-ACE, this is occurring via the TCA cycle and then through oxidation of NADH in the electron transport chain. Our data also indicate that lipids provide a sink for reductants in these conditions as has been described previously (Lee et al, 2013). However, our studies using deuterated cholesterol indicate that some reductant is also channelled into amino acid biosynthesis indicating that this provides Mtb with another potential sink for reductants.

The relatively large amounts of $CO_2$ produced from the CHL/ACE culture are in concordance with a partitioning of isocitrate through a complete TCA cycle and the glyoxylate shunt combined with gluconeogenic operation of PEPCK when growing on cholesterol and acetate. This mode of operation is in stark contrast to Mtb growing with GLY-OLA where we measured reduced flux through an incomplete TCA cycle and insignificant fluxes through the glyoxylate shunt combined with anaplerotic flux through PEPCK that leads to the lower $CO_2$ release in Mtb grown on GLY-OLA. OAA supply is in part satisfied by PEPCK functioning in the carboxylating direction, confirming our previous result (Beste et al, 2011), whilst gluconeogenesis proceeds directly from glycerol-derived PGA. Although MEZ, PCA (which are not distinguished in our model) and PEPCK are all able to fix carbon from $CO_2$ (Basu et al, 2018), PEPCK is predicted by these results to be the dominant enzyme of the anaplerotic node required for either gluconeogenesis or anaplerosis. This explains why Mtb strains lacking PEPCK are attenuated in a variety of different in vitro, ex vivo and in vivo conditions (Marrero et al, 2010; Baker & Abramovitch, 2018; Basu et al, 2018).

Previous studies have shown that Mtb strains lacking the bifunctional isocitrate lyase/methylisocitrate lyase (ICL) are unable to grow in the presence of cholesterol because they cannot metabolise propionate due to the role of this enzyme in the MCC (Lee et al, 2013). However, the addition of C2 containing metabolites alleviates this attenuation, by priming incorporation of propionyl-CoA into lipids and so limiting the build-up of toxic MCC intermediates and propionyl-CoA itself (Lee et al, 2013). Here, we show that even in the presence of an intact MCC, the flux of carbon from cholesterol into lipids is the preferred metabolic route when there are sufficient C2 metabolites to prime lipid biosynthesis as evidenced by the low flux through the MCC in the slow growth chemostat conditions. This was further confirmed by the lipid analysis which showed an increase in the incorporation of carbon from cholesterol into SL-1 resulting in an increase in chain length of this virulence lipid. Changes in SL-1 are interesting as this lipid plays a critical role in inducing the characteristic TB cough (Ruhl et al, 2020) thereby potentially linking clinical manifestations with changes in carbon source availability.

The role of the glyoxylate shunt in Mtb pathogenesis has always been obscured by the dual function of the gating enzyme ICL enzyme in both the shunt and the MCC. Enzymes which function

exclusively in the MCC are known to be dispensable in murine models of Mtb, prompting the hypothesis that vitamin B12-dependent methylmalonyl pathway (MMP) complements for the absence of the MCC pathway in vivo (Savvi et al, 2008). However, this is not consistent with the finding that strains lacking both a functional MCC and MMP but with a functioning glyoxylate shunt were significantly less attenuated in a murine model of TB than Δicl Mtb strains (Lotlikar, 2012). This indicates that in some conditions, ICL's only role is as the gateway enzyme into the glyoxylate shunt. Here, we show that when Mtb is growing slowly with cholesterol and fatty acids, there is very low flux through the MCC and the main function of ICL is to channel carbon through the glyoxylate shunt.

We previously identified an additional essential role for ICL during slow growth of Mtb on glycerol and Tween 80 using [13]C MFA, conditions in which the glyoxylate shunt would not be expected to be active (Beste et al, 2011). Here, using our extended isotopologue model, we demonstrate that under these conditions, ICL is catalysing a reverse flux through the MCC in the direction leading to propionyl-CoA generation. Reversal of the MCC has been described by Serafini et al, (2019) when Mtb is growing with pyruvate or lactate (Serafini et al, 2019). Our data suggest that this is a more general phenomena and may be active whenever there is a limited supply of odd-chain lipid precursors. The ability to operate the MCC as both a catabolic and an anabolic pathway provides Mtb with the metabolic flexibility to both detoxify excess sterol-derived propionate and, when necessary, to generate propionyl-CoA for lipid biosynthesis, including virulence-associated lipids. The generation of propionyl-CoA may be particularly important when Mtb is growing slowly and in stress conditions which are associated with increased demand for specific cell wall lipids (Singh et al, 2009; Warner, 2014). This also suggests that in conditions where a reversed MCC is required, the methylmalonyl pathway would be unable to complement the MCC pathway, highlighting the necessity for both pathways.

A role for a reversed MCC in the co-metabolism of glycerol and fatty acids is of interest in the context of the discovery that Mtb has access to glycerol in vivo (Safi et al, 2019). Although unlikely to be an essential carbon source, changes in glycerol metabolism have been shown to influence antibiotic tolerance of Mtb in vivo (Bellerose et al, 2019; Safi et al, 2019). Also, mutations in the MCC regulator prpR that conferred multidrug tolerance were identified in TB patients (Hicks et al, 2018). Although this finding was proposed to reflect an impaired ability of these strains to metabolise propionate, the results here suggest an alternative and intriguing hypothesis that dysregulation of the MCC may affect the metabolism of glycerol. Interestingly, Mtb strains lacking ICL were shown to be less tolerant to antibiotics in glycerol containing medium and this was associated with an increased flux through the TCA cycle (Nandakumar et al, 2014). We posit that this is due to the operation of a reversed MCC. When the MCC is reversed, methyl citrate lyase (ICL) becomes the gating enzyme into this cycle, and therefore, as Nandakumar et al eloquently demonstrated, ΔiclMtb strains would increase flux through the TCA cycle, altering the energetics of the cell and reducing Mtb tolerance to antibiotics (Nandakumar et al, 2014).

Surprisingly, our data predicted zero flux through the PPDK reaction presumed to be catalysed by the product of Rv1127c and has been shown by us, and others, to be essential for cholesterol and propionate metabolism (Griffin et al, 2011; Basu et al, 2018). PPDK's use two binding domains, for ATP/AMP and PEP/pyruvate,

to catalyse the reversible interconversion between pyruvate and PEP. Mysteriously, analysis of Rv1127c reveals that although the gene encodes an entire N-terminal PPDK domain to bind and dephosporylate ATP, the predicted protein completely lacks a C-terminal PEP/pyruvate-binding domain (Fig EV4), raising doubts over the annotated function of this protein. *Mycobacterium bovis, Mycobacterium bovis* BCG and *Mycobacterium leprae* encode similar "PPDK" genes lacking PEP/pyruvate-binding domains. Although it is possible that the mycobacterial PEP/pyruvate-binding domain is encoded on another Mtb gene, *in silico* searches have not identified any genes encoding Mtb proteins homologous to the C-terminal PEP/pyruvate-binding domain of typical PPDK's. Loci for functionally related genes tend to be in close genomic proximity in prokaryotes and the Rv1127c locus is nearby genes encoding enzymes and regulators of the methylcitrate cycle that, like Rv1127c, are essential for the metabolism of cholesterol and propionate. The absence of PEP/pyruvate-binding domain and, as we demonstrate in this study, zero flux through the PPDK reaction in conditions in which Rv1127c is essential prompts us to propose that Rv1127c is not a PPDK but performs an, as yet unknown, metabolic function associated with the methylcitrate cycle. This would be consistent with our previous data demonstrating propionate vulnerability of strains lacking Rv1127c that could be complemented by vitamin B12 to activate the methylmalonyl pathway. The precise function of Rv1127c may have therapeutic implications as ΔRv1127c is attenuated for intracellular survival and is more sensitive to the antibiotic bedaquiline (Mackenzie *et al*, 2020). Rv1127c is thereby an attractive target for developing anti-TB drugs which synergise with bedaquiline, a critical drug in the new shortened therapy against drug-resistant TB.

In conclusion, our results demonstrate that Mtb efficiently co-metabolises combinations of either glycerol or cholesterol along with C2 generating substrates to efficiently generate biomass. Although it has been suggested that Mtb is able to differentially co-metabolise carbon substrates to distinct metabolic fates, we find no evidence of compartmentalised metabolism in this study suggesting that this hypothesis may need re-evaluating. We show that a reversible MCC pathway along with the glyoxylate shunt provides Mtb with a high level of flexibility for re-routing metabolic fluxes depending on the combination of carbon sources being metabolised to generate biomass and in particular mycolipids. For example, we show that Mtb re-routes carbon flux through a reversed MCC when growing on glycerol and Tween 80. Finally, we find no evidence for PPDK flux under the conditions where it would be expected, indicating that Rv1127c is incorrectly annotated and that its actual role in the MCC remains unknown. Overall, this work illustrates the flexibility of Mtb metabolism, and how it is able to adapt to the different combinations of nutrients that this pathogen might encounter during infection. The work also highlights the importance of exploring metabolic flux profiles in different conditions if we wish to fully comprehend metabolic potential of Mtb.

# Materials and Methods

### Bacterial strains and growth conditions

*Mycobacterium tuberculosis* (H37Rv) was grown in a 2 litre bioreactor (Adaptive Biosystem Voyager) under aerobic conditions and at pH 6.6 as previously described (Beste *et al*, 2011). Chemostat cultures were grown in Roisin's minimal medium at a constant dilution rate of $0.01\ h^{-1}$ (slow growth rate). Culture samples were withdrawn from the chemostat to monitor cellular dry weight, viable counts, optical density and nutrient utilisation, and $CO_2$ and $O_2$ levels were measured in the exhaust gas (Beste *et al*, 2011). Metabolic steady state was confirmed from the $OD_{600}$, substrate consumption and $CO_2$ production rates.

Once metabolic steady state was reached, $^{13}C$ labelling experiments were initiated in the chemostat by replacing the feed medium with an identical medium containing a mixture of unlabelled and labelled carbon sources. Two mixtures of (i) 100% U-$[^{13}C_2]$ acetate and unlabelled cholesterol, or (ii) 30% U-$[^{13}C_3]$ glycerol and unlabelled Tween 80 were used as carbon substrates in Roisin's minimal media. To confirm isotopic steady state, cultures were withdrawn at regular intervals and the amount of $^{13}C$ label incorporation was measured using gas chromatography–mass spectrometry (GC-MS) analysis for every volume change.

To explore the potential flux through PPDK, Mtb was grown in Roisin's minimal medium containing 0.1% (vol/vol) $[3, 4-^{13}C_2]$ cholesterol as the sole carbon source until mid-log phase ($OD_{600} = 0.6$–0.8) before being harvested to analyse isotopologue incorporation into intracellular metabolites. To evaluate the incorporation of $^2H$ into the amino acids of Mtb, deuterated cholesterol (0.0625 g/l) was used as the single carbon source in Roisin's media. Deuterated cholesterol was prepared using a recombinant Pichia pastoris strain CBS7435 Δhis4Δku70 Δerg5::pPpGAP-ZeocinTM-[DHCR7] Δerg6::pGAP-G418[DHCR24] (54). Cultures were adapted to growth in deuterated minimal in the Deuteration Laboratory (D-Lab) within the Life Sciences of the Institut Laue-Langevin (Haertlein *et al*, 2016) and purified as described (Moulin *et al*, 2018). Mtb cultures grown in deuterated cholesterol were harvested at the mid-log phase for metabolomics.

### Culture analysis and substrate uptake measurements

Biomass and supernatant samples were collected and harvested (Beste *et al*, 2011). The amounts of glycerol in the supernatant and in fresh medium were assayed by use of a commercial assay kit that employs a glycerokinase-coupled enzyme assay system (Boehringer Mannheim). To assay for Tween 80, supernatant samples were hydrolysed by boiling for 1 h in methanolic KOH (5% potassium hydroxide: 50% methanol). After neutralising the samples with HCl, the free fatty acids were assayed using a commercial kit (Roche). Concentrations of cholesterol and acetate were measured using commercial kits supplied by Boehringer Mannheim.

### Biomass hydrolysate preparation and gas chromatography–mass spectrometry (GC-MS) analysis

Biomass samples from the $^{13}C$-labelled chemostat and mid-log batch cultures were washed and hydrolysed in 6 M Hydrochloric acid for 24 h (6). The dried hydrolysate was dissolved in 1 ml norvaline solution (0.075 mM in 80:20 H2O:MeOH). 0.1 ml was dried in-vacuo and this mixture was derivatised by adding 140 μl acetonitrile and N-tert-butyldimethylsilyl-N-methyltrifluoroacetamide (MTBSTFA), 1% tert-butyldimethylchlorosilane (TBDMCSI) (1:1) sonicating (room temperature, 30 min) and then heating (90°C, 30 min) to

complete the derivatisation. Samples were analysed by GC-MS within 72 h after derivatisation.

The analysis was performed with a GC system (Agilent) fitted with a DB-5ms capillary column (15 m × 0.18 mm internal diameter × 0.18 μm film with 5 m Duraguard integrated guard column) and deactivated quartz wool packed FocusLiner (SGE) coupled to a Pegasus III time-of-flight (TOF) mass spectrometer (Leco) equipped with a splitless injector. The injector temperature was initially held at 70°C for 2 min followed by heating to 350°C at a rate of 17°C/min. This temperature was held for 1.5 min. The He Carrier flow was held constant at 1.4 ml/min. The injection volume was 5 μl, and inlet temperature was 250 °C, interface temperature at 310 °C and source temperature at 245°C. The system was operated with a mass range of 40–800 $m/z$ at an EM voltage of 70 eV with a spectral acquisition rate of 20 spectra/s. $^{13}C$ isotopologue abundances (i.e. $^{13}C$ incorporation; $U-^{12}C$, $^{13}C_1$, …, $^{13}C_n$) for each amino acid were determined for fragments containing the intact carbon skeleton for each amino acid, generally using the $[M-57]^+$ ion. To obtain further information relating to the $^{13}C$ incorporation at individual positions within individual amino acids, isotopologue abundances were also determined for fragments formed by cleavage of the C1–C2 bond in some samples. Within a subset of these, isotopologue abundances for the fragment formed by cleavage of the C2–C3 bond in serine were also determined allowing comprehensive characterisation of the isotopic incorporation into serine. Mass spectra of the derivatised amino acids were corrected for the natural abundance of all stable isotopes (Beste *et al*, 2011; Borah *et al*, 2019).

### Sample preparation and liquid chromatography–mass spectrometry (LC-MS) analysis

For the detection of intracellular metabolites, cultures were quenched in a solution of 60% methanol and 0.9% NaCl at −20°C. The metabolites were extracted in a pre-chilled (−20°C) solution of methanol:chloroform (1:1), and cells were lysed twice by bead beating in the FastPrep at 6.5 for 20 s with careful cooling in between. Soluble extracts were filtered twice and then stored at −80°C until analysis by LC-MS.

Samples for $^{13}C$ isotopomer profiling of organic acids and sugar phosphates were separated using a synergy hydro C18 reversed phase column on an Agilent 1200 chromatography system (Agilent Technologies, Waldbronn, Germany). Free amino acids were separated using a Luna SCX cation exchange column on a JASCO HPLC system. In both cases, the HPLC outlets were coupled to an API 4000 (Applied Biosystems, Concord, Canada) mass spectrometer equipped with a TurboIon spray source. The separation methods, optimal MS parameter settings as well as extraction and integration of ion chromatograms as described (Kappelmann *et al*, 2017).

### MALDI lipid fingerprinting

A late log phase culture was washed in phosphate buffer saline by centrifugations and heat killed. The heat-inactivated was further washed three times in water and resuspended in 100 μl. An aliquot of 0.4 μl of the mycobacterial solution was loaded onto the target and immediately overlaid with 0.8 μl of 10 mg/ml super-2,5-dihydroxybenzoic acid (sDHB) matrix in chloroform/methanol (CHCl3/MeOH) 90:10 v/v. The mycobacterial sample and matrix were mixed directly on the target by pipetting and allowed to dry under a stream of air. MALDI-TOF MS analysis was performed on a 4800 Proteomics Analyzer (Applied Biosystems) using the reflectron mode. Samples were analysed by operating at 20 kV in the negative ion mode using an extraction delay time set at 20 ns.

### Metabolic modelling

Metabolic modelling was conducted using INCA, version 1.7 (Young, 2014). An extended isotopomer model of central metabolism (Table EV1) in Mtb was constructed of the model previously published (Beste *et al*, 2011). In addition to the reactions of glycolysis (EMP), gluconeogenesis, the pentose phosphate pathway (PPP), the tricarboxylic acid cycle (TCA) and anaplerosis (ANA), we included the reactions of the MCC and amino acid degradation pathways. This metabolic network model was supplemented with carbon atom transitions and consisted of a total of 90 reactions and 51 exchange reactions. Labelling measurements were obtained using 100% $[U-^{13}C_2]$ acetate for CHL-ACE and 30% $[U-^{13}C_3]$ glycerol for GLY-OLA. Flux values for net and exchange rates were derived from 86 independent flux parameters that are estimated using 108 labelling measurements for CHL-ACE and 120 for GLY-OLA, respectively (supplementary data file S1). Substrate consumption rates for cholesterol, acetate, glycerol and Tween 80 and overall $CO_2$ net production rate (Table 1) were used as fixed fluxes to constrain the model.

### Metabolic flux estimation

Flux estimations were performed using INCA (Young, 2014) using non-linear weighted least squares fitting approach to determine the flux values which are the most likely description of the labelling data and biomass constraints (Beste *et al*, 2011). INCA performs a Levenberg–Marquardt gradient-based search algorithm to minimise the sum of squared residuals (SSR) between the simulated and experimental measurements (Wiechert, 2001; Young, 2014). A multistart optimisation approach was used for flux estimations with a total number of restarts of 100. Flux distributions with a minimum statistically acceptable SSR were considered the best fit. The goodness of fit of the flux maps was assessed by comparing the simulated and experimental measurements. The upper and lower 95% confidence limits of the best-fit flux distributions were calculated using parameter optimisation option in INCA (Young, 2014). Fluxes were considered to be significantly different if the 95% confidence limits do not overlap.

## Data availability

The raw and processed mass isotopomer data and estimated fluxes are included as Tables EV2 and EV3. The isotopomer model is deposited on GitHub (https://github.com/KB-2021/TB-metabolic-model.git).

**Expanded View** for this article is available online.

## Acknowledgements

This work was supported by a grant from the Wellcome Trust (London) Grant Number (088677/Z/09/Z), Medical Research Council (MRC) grant reference (MR/K01224X/1), Biotechnology and Biological Sciences Research Council

(BBSRC) (BB/T007648/1) and National Institutes of Health (NIH) for financial support (P01-AI095208). The funders had no role in study, design, data collection and analysis, the decision to publish, or preparation of the manuscript.

## Author contributions

KB designed and conducted experiments, analysed data and wrote the manuscript. TM analysed data and wrote the manuscript. NDH conducted the experiments. JLW and MHB analysed data. GLM conducted experiments and analysed data. AB wrote the manuscript. MM, MH, GS, HP and VTF synthesised the deuterated hydrogen and designed experiments. SN analysed data. CWG analysed data. JM obtained funding and wrote manuscript. DJVB designed the experiment, obtained funding, analysed data and wrote manuscript.

## Conflict of interest

The authors declare that they have no conflict of interest.

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
