## [Review Process File · Molecular Systems Biology]

Metabolic fluxes for nutritional flexibility of *Mycobacterium tuberculosis*

Khushboo Borah, Tom Mendum, Nathaniel Hawkins, Jane Ward, Michael Beale, Gerald Larrouy-Maumus, Apoorva Bhatt, Martine Moulin, Michael Haertlein, Gernot Strohmeier, Harald Pichler, Trevor Forsyth, Stephan Noack, Celia Goulding, Johnjoe McFadden, and Dany Beste
DOI: 10.15252/msb.202110280

Corresponding author(s): Dany Beste (d.beste@surrey.ac.uk)

Review Timeline:

Submission Date:	8th Feb 21
Editorial Decision:	1st Mar 21
Revision Received:	17th Mar 21
Editorial Decision:	23rd Mar 21
Revision Received:	30th Mar 21
Accepted:	31st Mar 21

Editor: Maria Polychronidou

Transaction Report:

Thank you again for submitting your work to Molecular Systems Biology. We have now heard back from the three referees who agreed to evaluate your study. Overall, the reviewers acknowledge that the presented findings are interesting and potentially relevant for future drug target selection. They raise however a series of concerns, which we would ask you to address in a revision.

I think that the recommendations of the reviewers are rather clear. Therefore, I do not see the need to repeat the points listed below. All issues raised by the reviewers need to be satisfactorily addressed. Please contact me in case you would like to discuss in further detail any of the issues raised.

On a more editorial level, we would ask you to address the following points.

REFEREE REPORTS

Reviewer #1:

Review of MSB-2021-10280, Beste and co-authors entitled "Metabolic fluxes for nutritional flexibility of Mycobacterium tuberculosis."

The authors described metabolic flux analysis of MTB grown in their chemostat model using two different carbon sources thought to be relevant physiologically. The authors use C13 labeled precursors to monitor flux through central carbon metabolism and show that proteinogenic amino acids and intracellular metabolites have similar isotopic composition using both cholesterol-acetate

and glycerol-oleic acid medias. They then generate quantitative data regarding the isotopologue composition of the amino acids and utilize this data to refine a metabolic flux model under both conditions. Overall the work is excellent - I have a few minor concerns detailed below but this work is an important contribution to our understanding of exactly how MTB parses flux through these pathways and suggests some specific intervention points that may help guide future drug target selection.

Minor issues:

1. Line 175. Actually, for succinate it appears more like 80% of the label is coming from cholesterol, is this perhaps because there is just a lot more succinate produced in this condition?
2. Line 251. Probably I am confused but why is the pyruvate M+3 so abundant, the cholesterol source was only +2 so this seems weird, maybe I missed something here.
3. Line 296. Add to the discussion section the recent demonstration that SL-1 plays a role in inducing cough (PMID:32142653) and its abundance in CHL-ACE may be related to carbon source availability late in disease.
4. Line 296. This isn't unique to the sulfatides though, virtually every species is higher mass in the CHL-ACE media compared to the GLY-OLA media, this might just represent greater AcylCoA pools?
5. Line 328. Using perdeuterated cholesterol for the experiment shown in Figure 7B seems a bit problematic since every biosynthetic step that involves cleavage of a C-H/D bond would experience a primary deuterium isotope effect. Thus, the differential incorporation into branched chain amino acids may simply reflect the number of kinetic isotope effects experienced and not efficiency of incorporation. It still interesting but probably add a caveat (or contrast with another labeling study with another organism if that's known).
6. Line 376-7. "...in accordance a" to "in accordance with a"
7. Line 403. "dual" not "duel"
8. Figure 1. Left hand side GLY-Ola bar, fourth bar from top labeled as R5P. Should this be S7P? Unmentioned in the text so if there is a reason this was R5P it should be noted and explained in the legend.
9. Figure 6. Panel missing label "A"

Reviewer #2:

Summary

The manuscript by Borah and coworkers is an excellent continuation as well as a critical expansion of their previous studies, and it aims at an in-depth and quantitative description of Mtb metabolism. The study is truly at the forefront of Mtb research and provides a balanced top-notch experimental analysis of co-catabolism of substrates by Mtb, as well as an authoritative computational genome-

scale metabolic network analysis.

Using a previously published in vitro steady-state culture system in a chemically-defined medium, in a chemostat, the authors design elegant and conclusive experiments to trace a large number of isotopically-labelled substrates and metabolites. These rigorous experimental conditions allow them to revisit some of the published findings (by them and others), for example dealing with compartmentalised metabolism in Mtb and the detailed function and direction of various anaplerotic reactions and pathways.

The conclusions drawn appear extremely solid and well supported by the findings and could have a major impact on TB research and therapies. Beside confirming the role of important (essential) enzymes and pathways, the elegant co-catabolism experiments reveal that the previously proposed metabolic compartmentalisation was likely unfounded, and shed new light on the pivotal role of ICL in the glyoxylate shunt and not the MCC pathway in infection-like conditions.

General remarks and major points

Overall, the manuscript reads extremely well, and will interest a broad audience, not only Mtb aficionados. Now, that being said, I think the authors should take greater care to better explain some of their experimental choices and also should better place their findings in the big picture. Indeed, even though this study is the last one in a series of investigations, it should be more self-contained and the relevance for a broad audience made clearer in the discussion.

For example, the authors do not explain the reasons for the choice of the two "food combinations" CHL-ACE versus GLY-OLA. They simply refer to a previous article. I would like to know (and any reader would likely think the same) whether these diets are merely experimental convenience, or are they supposed to reflect diets encountered by Mtb during infection? Does it for example reflect the extracellular versus intracellular conditions?

In general, the authors have to make more explicit the correlations (or lack thereof) between their experimental conditions and the situations faced by Mtb during infection. The authors do not explicitly mention that Mtb can live outside cells (e.g. before uptake by alveolar macrophages, after release by dying macrophages, in the caseum), and inside various cell types (e.g. alveolar macrophages, interstitial macrophages, neutrophils, epithelial and endothelial cells, and even more complex, in M1 versus M2 macrophages). Finally, and quite importantly, even during its intracellular life, Mtb spends time in a (damaged) vacuole of phagosomal origin, but also spends time in the cytosol. Therefore, simple questions that need to be addressed explicitly for the sake of non-specialists are whether the four carbon sources (CHL, ACE, GLY, OLA) are available in all cells and compartments? And even more narrowly, are the combinations CHL-ACE and GLY-OLA making sense at all? One could imagine that CHL is membrane-associated, while the concentration of OLA as a free fatty acid is almost zero and therefore has to be first obtained by hydrolysis of TAG and phospholipids. I am not a specialist, but there is likely no ACE and GLY in a vacuole, and even in the cytosol, their concentrations are likely very low and they also have to be obtained from other metabolites (Acetyl-CoA etc...).

In a similar manner, the defined medium chosen does not have any vitamin (especially B12) ... how does that reflect (or not) accessibility to vitamins inside a host cell?

Finally, the author describe their chemostat conditions as "highly oxygenated" ... how does that reflect (or not) the conditions faced by Mtb, inside and outside the various compartments and cells

mentioned above, as well as in the granuloma?

I am convinced that, if the authors go that extra length, the significance and relevance of their solid and elegant findings will be dramatically increased.

Reviewer #3:

This manuscript seeks to define Mtb metabolic flexibility by elucidating the metabolic fluxes through the Mtb's central carbon metabolic pathways unique to cocatabolize multiple carbon sources (1. CHL/ACE and 2. GLY/OLA). Authors also identified that Mtb metabolic flexibility doesn't rely on metabolic compartmentalization, unlike the previous studies. The authors use ¹³C MFA and identified that Mtb use full set of TCA/Glyoxylate shunt with no significant contribution of MCC when using CHL/ACE but use TCA and reverse MCC to produce toxic lipid intermediates when using GLY/OLA.

Also, authors claimed Mtb induces MCC activity in a reverse direction when exposed to GLY/OLA carbons due to shortage of odd chain fatty acids required for the synthesis of cell wall lipid constructs. ¹³C MFA results also showed no carbon flux from pyruvate to PEP encoded by Rv1127c and with this, authors proposed that Rv1127c is not functional PPDK but involved in MCC activity.

The experiments were done well and there is a lot of useful information in this study regarding conceptually novel Mtb metabolic flexibility that presumably serves as a groundwork to propose novel TB drug targets. Although the manuscript is interesting, numerous items need to be addressed before it is suitable for publication.

First, authors claimed based upon the results of the ¹³C MFA that Mtb catabolize multiple carbons with no evidence of compartmentalization. However, authors have revealed that Mtb when exposed to CHL/¹³C ACE, most of CHL was used to biosynthesize cell wall lipids (Fig. 6) while consuming most of ACE through intact TCA and glyoxylate shunt. Also, when exposed to GLY/OLA, GLY is preferentially metabolized through glycolytic/incomplete TCA cycle, while activating reversed MCC using pyruvate and succinate to produce odd chain fatty acid CoA for the precursors of cell wall lipids. If I understand correctly, this should be the evidence of metabolic compartmentalization, whereby each carbon has its own metabolic fate. Also, authors claimed that glycerol and CHL are two important carbons available inside hosts. Then, ¹³C MFA of Mtb when cultured in CHL and ¹³C GLY should be more informative to understand the metabolic flexibility to adapt host carbon condition if authors describe metabolic fluxes when exposed to CHL and GLY as compared to those of CHL or GLY only. Because they have distinct entry points within Mtb central carbon metabolism.

Another evidence the authors raised showing no compartmentalization included that most of proteinous AA showed uniform ¹²C and ¹³C fractions. AA biosynthesis pathways are linked to Mtb CCM but catalytically positioned at the same direction used to consume CHL and ACE or GLY and OLA as substrates. Thus, it is somewhat difficult to conclude that Mtb has no compartmentalized metabolic pattern when multiple carbons are used.

Also, ¹³C labeling rates of PYR, SUC, VAL, LEU and ALA were less than 50%, indicating the major contribution of CHL rather than ACE on their biosynthesis (Fig. 1). Although PYR is arising from CHL degradation, branched AA labeling rates resemble PYR but chorismate metabolism aromatic AA showed greater than 50% labeling rate. Thus, PYR arising from CHL serves as a preferred

substrate of branched AA rather than that of aromatic AA; this is also metabolic compartmentalization. DHAP also showed greater than 50% labeling rate, indicating ACE mediated acetyl-CoA serves as a preferred substrate of gluconeogenesis but acetyl-CoA/pyruvate from CHL serve as substrate for branched AA.

Fig. 1: In culture using 30% ¹³C GLY+70% ¹²C GLY + 100% ¹²C OLA, how labeling of metabolites from ¹²C fraction of GLY and from ¹²C OLA was distinguished ?

Line 187: isoleucine (ILE).

¹³C isotopologue analysis to define the synthesis origin provides the results as expected and not surprising.

Fig. 4: some enzyme net flux was 0. Does this have no carbon flux ?

Line 227: MFA of Mtb under CHL+ACE showed that canonical TCA cycle is active; authors concluded that Mtb uses an incomplete TCA cycle... These two sentences were conflict.

Lines 231 - 236: It is a very interesting finding. PEPCK catalytic direction is affected by redox state and ATP level in response to environmental changes. Thus, authors can monitor the NADH/NAD ratio and ATP/GTP levels of Mtb under CHL/ACE or GLY/OLA to validate the metabolic bases underlying the net flux through PEPCK. Also, as an option, PEPCK deficient Mtb can be used to confirm the functional essentiality of PEPCK under the culture conditions.

Line 236: ANA R2 overall flux is low but the direction is opposite and greater when cultured in GLY-OLA. Data figure and interpretation are not matched.

Line 244: reference.

Line 246: PPK requirement in Mtb cultured in CHL can be validated by conducting MFA of Mtb cultured in only CHL. The experiment can support the metabolic remodeling of Mtb required to cocatabolize multiple carbons as compared to that catabolize single carbon source.

Lines 250 - 254 and Fig. 4E : Figure 4E showed around 10% M1 labeled in PEP and around 30% M3 labeled PYR when Mtb cultured in ¹³C₂ CHL media. In sentence, they explained M2 PYR was identified with M1 PEP. Figure and explanation were not matched. Also, authors mentioned all PEP were all M1 but in figure, 70% PEP were ¹²C fraction.

Line 255, Fig. 4C, D: Glycolytic/gluconeogenic carbon fluxes in both directions were almost shut down. Does this mean their catalytic activities are not essential ? Authors may check the mRNA expression of these genes in Mtb after culturing in CHL/ACE or GLY/OLA condition. Authors also can check the essentiality of the genes under either condition.

Lines 271 - 273: Authors need to compare MFA through MCC from CHL + ACE as compared to that of CHL only. If MCC is active in Mtb when culturing in the presence of CHL only, the altered MCC upon addition of ACE indicated catalytic compartmentalization by which CHL mediated propionyl-CoA is preferentially metabolized for the lipid biosynthesis and ACE mediated acetyl-CoA is metabolized for the TCA cycle intermediates. Thus, this finding, if combined with MFA result of CHL single carbon, also provide the evidence of catalytic compartmentalization when cocatabolizing multiple carbons.

Line 278: What does it mean by extracellular biomass in Table 1 ?

Fig. 6: "A" is missing.

Fig. 6:

Line 304: CHO typo

Line 314: through

Reviewer 1

1. Line 175. Actually, for succinate it appears more like 80% of the label is coming from cholesterol, is this perhaps because there is just a lot more succinate produced in this condition?

We haven't measured pool sizes in this study as they don't correlate with metabolic fluxes. It is of course possible that more succinate is excreted or produced under these conditions. As CHL enters central carbon metabolism as succinate it is expected that more of this intermediate will be derived from cholesterol. See the text below in the manuscript.

L174-178: *For the CHL-ACE experiments, the labelling profile of succinate (SUC), pyruvate (PYR), and the pyruvate derived amino acids (alanine (ALA), valine (VAL) and leucine (LEU)) had $\geq 50\%$ unlabelled carbon indicating that the carbon backbone of these metabolites was predominantly derived from unlabelled cholesterol. This is expected as cholesterol enters central carbon metabolism as succinate and pyruvate.*

2. Line 251. Probably I am confused but why is the pyruvate M+3 so abundant, the cholesterol source was only +2 so this seems weird, maybe I missed something here.

The reviewer is completely right and Figure was plotted incorrectly. We apologise for our error and have now replotted and included the pyruvate derived amino acid alanine. The figure is now in concordance with the text in the manuscript.

3. Line 296. Added to the discussion section the recent demonstration that SL-1 plays a role in inducing cough (PMID:32142653) and its abundance in CHL-ACE may be related to carbon source availability late in disease.

We have added the following to the manuscript (L407-409).

L404-406: *Changes in SL-1 are interesting as this lipid plays a critical role in inducing the characteristic TB cough (PMID:32142653) thereby potentially linking clinical manifestations with changes in carbon source availability.*

4. Line 296. This isn't unique to the sulfatides though, virtually every species is higher mass in the CHL-ACE media compared to the GLY-OLA media, this might just represent greater AcylCoA pools?

The higher mass observed for the other species was not seen in the unlabelled control and therefore is only due to the extra mass from the ^{13}C label in acetate as explained in the exert from the manuscript below (L300-307).

L300-307: *We also found significant mass shifts in triacylphosphatidylinositol dimannosides (AcPIMs) isolated from the chemostat samples. However, these shifts were not observed in the control samples indicating that the in this case the increased mass was due to ^{13}C labelled acetate or glycerol being incorporated into these lipids. For CHO-ACE-grown Mtb, the observed increased mass shift (Fig. S3A, B) was likely due to incorporation of [$^{13}\text{C}_2$] acetate) directly into these lipids in contrast to the GLY-OLA chemostat grown Mtb where unlabelled OLA was being predominantly used for acetylation (Fig. S3).*

5. Line 328. Using perdeuterated cholesterol for the experiment shown in Figure 7B seems a bit problematic since every biosynthetic step that involves cleavage of a C-H/D bond would experience a primary deuterium isotope effect. Thus, the differential incorporation into branched chain amino acids may simply reflect the number of kinetic isotope effects experienced and not efficiency of incorporation. It still interesting but probably add a caveat (or contrast with another labelling study with another organism if that's known).

We agree that the possibility of an isotope effect cannot be totally excluded and therefore have added the following text and references to the manuscript (L327-331).

L327-331: *Interpretation of ^2H labelling data is complicated by the presence of significant deuterium kinetic isotope effects (<https://doi.org/10.1038/nchembio.2047>). However, ^2H tracers have proved invaluable in resolving redox metabolism in different organisms <https://doi.org/10.1016/j.molcel.2014.05.008>, <https://doi.org/10.1016/j.cell.2018.03.055>).*

6. Line 376-7. "...in accordance a" to "in accordance with a"

Corrected

7. Line 403. "dual" not "duel"

Corrected

8. Figure 1. Left hand side GLY-Ola bar, fourth bar from top labeled as R5P. Should this be S7P? Unmentioned in the text so if there is a reason this was R5P it should be noted and explained in the legend.

S7P was measured for CHL-ACE and R5P was measured for GLY-OLA cultures. We have made it clear in the legend for Fig. 1

Fig. 1. ¹³C incorporation into the proteinogenic amino acids and intracellular metabolites from metabolic and isotopic steady state chemostat Mtb cultures. Label distribution is shown in metabolites from Mtb grown in 30% [¹³C₃] glycerol and unlabelled Tween80 which provides Mtb with oleic acid (GLY-OLA) and 100% [¹³C₂] acetate and unlabelled cholesterol (CHL-ACE). Average ¹³C incorporation was calculated for metabolites harvested at a steady state growth and are shown as the amount labelled (¹³C) and unlabelled (¹²C). Data are average of 3 replicate measurements. DHAP (dihydroxyacetone phosphate), PYR (pyruvate) ALA (alanine), GLY (glycine), SER (serine), LYS (lysine), MET (methionine), ASP/N (aspartate/asparagine), THR (threonine), ILE (isoleucine), ORN (ornithine), GLU/N (glutamate/glutamine), SUC (succinate), LEU (leucine), ILE (isoleucine), VAL (valine), PHE (phenylalanine), TYR (tyrosine), HIS (histidine), **S7P (sedoheptulose-7-phosphate), measured for CHL-ACE and R5P (ribose 5-phosphate), measured for GLY-OLA cultures** are plotted on a metabolic map showing reactions for glycolysis, PPP, GLX (glyoxylate shunt) and the TCA cycle. Aspartate/asparagine and glutamate/glutamine pools are lumped as both asparagine and glutamine were reduced to aspartate and glutamate respectively during acid hydrolysis.

9. Figure 6. Panel missing label "A"

Response: We have labelled "A"

Reviewer 2

1. I think the authors should take greater care to better explain some of their experimental choices and also should better place their findings in the big picture. Indeed, even though this study is the last one in a series of investigations, it should be more self-contained and the relevance for a broad audience made clearer in the discussion. For example, the authors do not explain the reasons for the choice of the two "food combinations" CHL-ACE versus GLY-OLA. They simply refer to a previous article. I would like to know (and any reader would likely think the same) whether these diets are merely experimental convenience, or are they supposed to reflect diets encountered by Mtb during infection? Does it for example reflect the extracellular versus intracellular conditions? In general, the authors have to make more explicit the correlations (or lack thereof) between their experimental conditions and the situations faced by Mtb during infection. The authors do not explicitly mention that Mtb can live outside cells (e.g. before uptake by alveolar macrophages, after release by dying macrophages, in the caseum), and inside various cell types (e.g. alveolar macrophages, interstitial macrophages, neutrophils, epithelial and endothelial cells, and even more complex, in M1 versus M2 macrophages). Finally, and quite importantly, even during its intracellular life, Mtb spends time in a (damaged) vacuole of phagosomal origin, but also spends time in the cytosol. Therefore, simple questions that need to be addressed explicitly for the sake of non-specialists are whether the four carbon sources (CHL, ACE, GLY, OLA) are available in all cells and compartments? And even more narrowly, are the combinations CHL-ACE and GLY-OLA making sense at all? One could imagine that CHL is membrane-associated, while the concentration of OLA as a free fatty acid is almost zero and therefore has to be first obtained by hydrolysis of TAG and phospholipids. I am not a specialist, but there is likely no ACE and GLY in a vacuole, and even in the cytosol,

their concentrations are likely very low and they also have to be obtained from other metabolites (Acetyl-CoA etc...).

In this study we are not trying to mimic the extremely complex *in vivo* conditions. Rather we are using the highly defined environmental conditions of steady state chemostat to explore the metabolic flux profile of Mtb metabolising physiologically relevant carbon sources. We have discussed the relevance of our selection of carbon sources in the introduction and to further clarify the context we have added the following to the introduction.

L56-61: *Mtb spends much of its life cycle growing intracellularly within the phagosomal compartment of macrophages where nutrient availability will fluctuate (doi: 10.1084/jem.20172020). In addition to replicating within the phagosomal compartment, it has been shown that Mtb can escape the intracellular environment to survive extracellularly (doi: 10.1128/aac.47.3.833-836.2003), in other cell types and within the diverse and dynamic microenvironments of granulomas (doi: 10.1093/femsre/fuz006).*

And also

L108-118: *During infection it is thought that Mtb uses fatty acids to prime this process further suggesting that cholesterol and fatty acid metabolism occurs simultaneously in vivo (19, 24-26). Despite numerous biochemical studies to elucidate the biochemical degradation pathways and studies exploring the role of specific enzymes in cholesterol and fatty acid metabolism the metabolic flux profile of Mtb growing on this combination of substrates has never been directly measured. Therefore in this study we performed ¹³C-MFA on steady state, slowly growing cultures of Mtb using an extended version of our ¹³C isotopomer model (4), which includes the MCC. We compared the metabolic flux profile of chemostat cultures of Mtb growing in carbon limited conditions growing with cholesterol/acetate with those growing on glycerol/Tween80 (oleic acid)."*

Line 113-118: *Therefore in this study we performed ¹³C-MFA on steady state, slowly growing cultures of Mtb using an extended version of our ¹³C isotopomer model (4), which includes the MCC. We compared the metabolic flux profile of chemostat cultures of Mtb growing in defined carbon limited conditions growing with cholesterol/acetate with those growing on glycerol/Tween80 to reflect carbon sources available to Mtb during the lifecycle of this pathogen within the host.*

And in the discussion we have added.

L333-335: *Mtb is known to be able to acquire and assimilate multiple carbon sources from the host during infection including sugars, host-derived fatty acids, amino acids, and cholesterol as sources of nitrogen, carbon, and energy during host infection.*

1. In a similar manner, the defined medium chosen does not have any vitamin (especially B12) ... how does that reflect (or not) accessibility to vitamins inside a host cell?

Our media does not contain vitamin B12 because this would activate the methyl malonyl pathway which is currently not present in our already large ¹³C isotopomer model of central carbon metabolism. We do not claim to be mimicking the *in vivo* situation. However regarding the availability of B12 to Mtb in the human host this is currently not established. A possible transporter of vitamin B12 has been reported but there remains no direct evidence of Mtb scavenging vitamin B12 from its intracellular niche.

2. Finally, the author describe their chemostat conditions as "highly oxygenated" ... how does that reflect (or not) the conditions faced by Mtb, inside and outside the various compartments and cells mentioned above, as well as in the granuloma?

It is likely that Mtb survives both aerobically and in caseous granulomas which may have reduced oxygen levels. However modelling metabolism of Mtb in reduced oxygen levels is beyond the scope of this current study.

Reviewer 3

1. Authors claimed based upon the results of the ^{13}C MFA that Mtb catabolize multiple carbons with no evidence of compartmentalization. However, authors have revealed that Mtb when exposed to CHL/ ^{13}C ACE, most of CHL was used to biosynthesize cell wall lipids (Fig. 6) while consuming most of ACE through intact TCA and glyoxylate shunt. Also, when exposed to GLY/OLA, GLY is preferentially metabolized through glycolytic/incomplete TCA cycle, while activating reversed MCC using pyruvate and succinate to produce odd chain fatty acid CoA for the precursors of cell wall lipids. If I understand correctly, this should be the evidence of metabolic compartmentalization, whereby each carbon has its own metabolic fate.

The metabolic flux profile is of the combined substrates. The propionate derived from cholesterol is directly incorporated into lipids. This isn't compartmentalised metabolism as this source doesn't actually enter central carbon metabolism. Indeed if you look at Figure 1 you can see uniform distribution of the label within central carbon metabolism showing no evidence of compartmentalisation as discussed (see also above comments to reviewer 1 about succinate). Similarly, we do not show that the glycerol preferentially enters central carbon metabolism. The flux results are for BOTH carbon sources.

2. Also, authors claimed that glycerol and CHL are two important carbons available inside hosts. Then, ^{13}C MFA of Mtb when cultured in CHL and ^{13}C GLY should be more informative to understand the metabolic flexibility to adapt host carbon condition if authors describe metabolic fluxes when exposed to CHL and GLY as compared to those of CHL or GLY only. Because they have distinct entry points within Mtb central carbon metabolism.

Experimental evidence indicates that cholesterol and lipids are present in the *in vivo* environment as discussed in the introduction. The glycerol/Tween 80 experiments almost acts as a control experiment as we have previously determined the metabolic flux under these conditions. Performing experiments using lots of different combinations of carbon substrates would indeed be interesting but will need to be the subject of future studies.

3. Another evidence the authors raised showing no compartmentalization included that most of proteinous AA showed uniform ^{12}C and ^{13}C fractions. AA biosynthesis pathways are linked to Mtb CCM but catalytically positioned at the same direction used to consume CHL and ACE or GLY and OLA as substrates. Thus, it is somewhat difficult to conclude that Mtb has no compartmentalized metabolic pattern when multiple carbons are used.

Amino acids are routinely measured instead of the intermediary metabolites for ^{13}C -Metabolic Flux Analysis as the amino acids in cell extracts and cell protein are much more abundant and stable than their precursors and provide extensive labelling information. Knowing the precursor-amino acid relationships it is easy to deduce the labelling patterns of the precursor metabolites from labelling patterns of the amino

acids. The ¹³C label incorporation in the mass isotopomers of proteinogenic amino acids retro-biosynthetically report on the central metabolite precursors and corresponding pathways. We have added to the manuscript the below text to clarify this.

L152-158: *Proteogenic amino acids are commonly measured for ¹³C-MFA as they are much more abundant and stable than their precursors and provide extensive labelling information. Moreover, amino acids can be used to directly deduce the labelling patterns of their precursor metabolites.*

4. Also, ¹³C labeling rates of PYR, SUC, VAL, LEU and ALA were less than 50%, indicating the major contribution of CHL rather than ACE on their biosynthesis (Fig. 1). Although PYR is arising from CHL degradation, branched AA labeling rates resemble PYR but chorismate metabolism aromatic AA showed greater than 50% labeling rate. Thus, PYR arising from CHL serves as a preferred substrate of branched AA rather than that of aromatic AA; this is also metabolic compartmentalization. DHAP also showed greater than 50% labeling rate, indicating ACE mediated acetyl-CoA serves as a preferred substrate of gluconeogenesis but acetyl-CoA/pyruvate from CHL serve as substrate for branched AA.

Pyruvate, succinate, valine and alanine (valine, alanine and leucine are derived from pyruvate) are the entry points of cholesterol into central metabolism and therefore would be expected to be derived more from cholesterol. However, there is no evidence of any compartmentalisation of flux through central carbon metabolism as evidenced by expected labelling of the amino acids produced from these pathways. We agree there is a small increase in ¹³C labelling of DHAP however as this is not supported by any changes in the amino acids synthesised from the other precursors of glycolysis this does not support the conclusion that there is any compartmentalisation occurring here.

WE have added the text below to clarify this:

L174-184: *For the CHL-ACE experiments, the labelling profile of succinate (SUC), pyruvate (PYR), and the pyruvate derived amino acids (alanine (ALA), valine (VAL) and leucine (LEU)) had \geq 50% unlabelled carbon indicating that the carbon backbone of these metabolites was predominantly derived from unlabelled cholesterol. This is expected as cholesterol enters central carbon metabolism as succinate and pyruvate. Canonical ¹³C labelling patterns were measured in the other metabolites reflecting that 60% of the total carbon was derived from ¹³C labelled acetate and the remainder derived from unlabelled cholesterol indicating that metabolism was also not compartmentalised in these conditions (Fig. 1). For GLY-OLA grown cultures, the labelling profiles was consistent across the different metabolites analysed; the backbone of these metabolites was synthesized primarily from glycerol, demonstrating that metabolism of GLY-OLA was also not compartmentalised.*

5. Fig. 1: In culture using 30% ¹³C GLY+70% ¹²C GLY + 100% ¹²C OLA, how labeling of metabolites from ¹²C fraction of GLY and from ¹²C OLA was distinguished ?

We can determine how much GLY is C12 and C13 as we know the total amount of ¹³C label and the total amount of carbon from glycerol in the medium. Our data shows that 30% of our amino acids are as expected labelled and therefore we can calculate how much of the remaining glycerol is C12.

Line 187: isoleucine (ILE).

Corrected

6. ¹³C isotopologue analysis to define the synthesis origin provides the results as expected and not surprising.

Im not sure what the reviewer is referring to here?

7. Fig. 4: some enzyme net flux was 0. Does this have no carbon flux ?

None of the fluxes are zero they are just very small.

8. Line 227: MFA of Mtb under CHL+ACE showed that canonical TCA cycle is active; authors concluded that Mtb uses an incomplete TCA cycle... These two sentences were conflict.

Mtb uses an incomplete TCA cycle when growing on GLYC/OAA as stated in the manuscript. The canonical cycle was used when growing with CHL/ACE.

9. Lines 231 - 236: It is a very interesting finding. PEPCK catalytic direction is affected by redox state and ATP level in response to environmental changes. Thus, authors can monitor the NADH/NAD ratio and ATP/GTP levels of Mtb under CHL/ACE or GLY/OLA to validate the metabolic bases underlying the net flux through PEPCK. Also, as an option, PEPCK deficient Mtb can be used to confirm the functional essentiality of PEPCK under the culture conditions.

We agree this is interesting but measuring the redox state is beyond this current study. The fact that there is flux though PEPCK would not automatically mean that this enzyme is essential. However, as referenced in the manuscript we have previously shown in Basu et al (2018) that PEPCK is indeed essential for growth on CHL/ACE.

10. Line 236: ANA R2 overall flux is low but the direction is opposite and greater when cultured in GLY-OLA. Data figure and interpretation are not matched.

The figure shows as stated in the text that L238-239 "The overall flux through ANA R2 is very low and not significantly different when growing on CHL-ACE or GLY-OLA." The reviewer must be looking at the data for R1?

11. Line 244: reference.

Added

12. Line 246: PPDK requirement in Mtb cultured in CHL can be validated by conducting MFA of Mtb cultured in only CHL. The experiment can support the metabolic remodeling of Mtb required to cocatabolize multiple carbons as compared to that catabolize single carbon source.

This experiment is beyond the current study as would be a whole new study focused on metabolism of sole carbon sources. We don't feel that this lengthy experiment would add anything to the current study or add anything significant to our conclusions.

13. Lines 250 - 254 and Fig. 4E: Figure 4E showed around 10% M1 labeled in PEP and around 30% M3 labeled PYR when Mtb cultured in ¹³C2 CHL media. In

sentence, they explained M2 PYR was identified with M1 PEP. Figure and explanation were not matched. Also, authors mentioned all PEP were all M1 but in figure, 70% PEP were 12C fraction.

This figure has now been corrected. Apologies for the error.

14. Line 255, Fig. 4C, D: Glycolytic/gluconeogenic carbon fluxes in both directions were almost shut down. Does this mean their catalytic activities are not essential ? Authors may check the mRNA expression of these genes in Mtb after culturing in CHL/ACE or GLY/OLA condition. Authors also can check the essentiality of the genes under either condition.

Metabolism is very robust and therefore this doesn't necessarily indicate that the catalytic activities are essential and therefore essentiality studies are not warranted here. Expression of metabolic genes does not reliably relate to metabolic fluxes as these enzymes are commonly regulated post-transcriptionally. I therefore don't think that either transcriptomic or essentiality analysis would add anything to our main conclusions and findings.

15. Lines 271 - 273: Authors need to compare MFA through MCC from CHL + ACE as compared to that of CHL only. If MCC is active in Mtb when culturing in the presence of CHL only, the altered MCC upon addition of ACE indicated catalytic compartmentalization by which CHL mediated propionyl-CoA is preferentially metabolized for the lipid biosynthesis and ACE mediated acetyl-CoA is metabolized for the TCA cycle intermediates. Thus, this finding, if combined with MFA result of CHL single carbon, also provide the evidence of catalytic compartmentalization when cocatabolizing multiple carbons.

Please see comments above. This study is focused on co-metabolism. Repeating everything on single carbon sources would be very interesting but these very time-consuming experiments await future studies.

16. Line 278: What does it mean by extracellular biomass in Table 1 ?

We are just referring to the biomass and therefore have removed extracellular

17. Fig. 6: "A" is missing.

Corrected

18. Fig. 6:

Corrected

19. Line 304: CHO typo

Corrected

20. Line 314: through

Corrected

Editorial level edits

-The keywords need to be reduced to 5.

The keywords are reduced to 5:

Mycobacterium tuberculosis, Tuberculosis, metabolism, metabolic flux, chemostat

- Please provide a .doc file for the manuscript text (including legends for the main figures and EV Figures) and individual production-quality files for the main figures and EV Figures (one file per figure).

Modified EV figures and legends in the revised submission

- We have replaced Supplementary Information by the Expanded View (EV format) in which a limited number of additional Figures (typically ~5) are included in the article as EV figures. In this case (unless during revision the number of additional figures becomes > 5) all additional figures can be displayed as EV Figures. EV Figures should be provided as individual files and their legends should be included in the main text. For detailed instructions regarding expanded view please refer to our Author Guidelines: <http://msb.embopress.org/authorguide#expandedview>.

Please see response above

- Please provide a "standfirst text" summarizing the study in one or two sentences (approximately 250 characters), three to four "bullet points" highlighting the main findings and a "synopsis image" (550px width and max 400px height, jpeg format) to highlight the paper on our homepage.

Standfirst text

This study applied quantitative metabolic analysis using stable isotopes, lipid fingerprinting and mathematical modelling to investigate the metabolic network of *Mycobacterium tuberculosis* growing slowly in a steady state chemostat system.

- We demonstrate that the tubercle bacillus efficiently co-metabolises either cholesterol or glycerol, in combination with two-carbon generating substrates without the need for compartmentalisation of metabolism.
- Metabolic flux profiles of *Mycobacterium tuberculosis* growing slowly on the dual carbon sources were described using an expanded ¹³C isotopomer model, which included the methyl citrate cycle.
- Partitioning of metabolite flux between the TCA cycle and the glyoxylate shunt combined with a reversible methyl citrate cycle were identified as critical metabolic nodes that underlie the metabolic flexibility of *M. tuberculosis*.

Figure:

Synopsis provided with the submission

- Table S1 should be provided as Table EV1. Please provide it as an .xls file. The description of the Table should be included in a separate tab in the .xls file.

We have now included Table S1 as Table EV1 .xls file

- The References should be formatted according to the Molecular Systems Biology reference style.

Completed

- Please include a Data availability section describing how the data and model have been made available. This section needs to be formatted according to the example below: The datasets and computer code produced in this study are available in the following databases: - Chip-Seq data: Gene Expression Omnibus GSE46748 (<https://www.ncbi.nlm.nih.gov/geo/query/acc.cgi?acc=GSE46748>) - Modeling computer scripts: GitHub (<https://github.com/SysBioChalmers/GECKO/releases/tag/v1.0>)

- [data type]: [full name of the resource] [accession number/identifier] ([doi or URL or identifiers.org/DATABASE:ACCESSION]) - The model should be provided as Computer Code EV1 or should be deposited on GitHub.

We have included a data availability statement in the revised manuscript

The analysed mass isotopomer data are included as excel file (File EV2). The isotopomer model is deposited on github (<https://github.com/KB-2021/TB-metabolic-model.git>).

- For data quantification: please specify the name of the statistical test used to generate error bars and P values, the number (n) of independent experiments

(specify technical or biological replicates) underlying each data point and the test used to calculate p-values in each figure legend. The figure legends should contain a basic description of n, P and the test applied. Graphs must include a description of the bars and the error bars (s.d., s.e.m.).

We have now provided this information.

Thank you again for sending us your revised manuscript. We think that the performed revisions have satisfactorily addressed the reviewers' concerns. Therefore I am glad to inform you that we can soon accept your study for publication in Molecular Systems Biology, pending some minor issues listed below.

Specifically, we would ask you to address the following editorial issues:

2nd Authors' Response to Reviewers**30th Mar 2021**

The authors have made all requested editorial changes.

Accepted**31st Mar 2021**

Thank you again for sending us your revised manuscript. We are now satisfied with the modifications made and I am pleased to inform you that your paper has been accepted for publication.

Corresponding Author Name: Dany Beste

Manuscript Number: MSB-2021-10280